# NDP52 mediates an antiviral response to hepatitis B virus infection through Rab9-dependent lysosomal degradation pathway

Shuzhi Cui[1], Tian Xia[1,2], Jianjin Zhao ⓘ[1], Xiaoyu Ren[1], Tingtao Wu[1], Mireille Kameni[1], Xiaoju Guo[1], Li He[1], Jingao Guo[1], Aléria Duperray-Susini[2], Florence Levillayer[2], Jean-Marc Collard ⓘ[1,2], Jin Zhong[1], Lifeng Pan ⓘ[3], Frédéric Tangy ⓘ[2], Pierre-Olivier Vidalain ⓘ[2,4], Dongming Zhou ⓘ[5], Yaming Jiu ⓘ[1], Mathias Faure ⓘ[4] & Yu Wei ⓘ[1,2] ✉

Autophagy receptor NDP52 triggers bacterial autophagy against infection. However, the ability of NDP52 to protect against viral infection has not been established. We show that NDP52 binds to envelope proteins of hepatitis B virus (HBV) and triggers a degradation process that promotes HBV clearance. Inactivating NDP52 in hepatocytes results in decreased targeting of viral envelopes in the lysosome and increased levels of viral replication. NDP52 inhibits HBV at both viral entry and late replication stages. In contrast to NDP52-mediated bacterial autophagy, lysosomal degradation of HBV envelopes is independent of galectin 8 and ATG5. NDP52 forms complex with Rab9 and viral envelope proteins and links HBV to Rab9-dependent lysosomal degradation pathway. These findings reveal that NDP52 acts as a sensor for HBV infection, which mediates a unique antiviral response to eliminate the virus. This work also suggests direct roles for autophagy receptors in other lysosomal degradation pathways than canonical autophagy.

The nuclear dot protein 52 (NDP52) belongs to a class of autophagy receptors called sequestosome 1/p62-like receptors[1,2]. NDP52 functions broadly in autophagy, not only in substrate recognition, but also in autophagy initiation and autophagosome maturation[3–7]. Autophagy is a lysosomal degradation pathway that plays an important part in host defense[8,9]. NDP52 orchestrates diverse cellular responses to pathogen invasion which are highly context-specific. NDP52 mediates selective autophagy of bacteria through ATG5-dependent canonical pathway[7,10–12]. In viral infection, NDP52 can target MAVS for autophagic degradation, suppressing type-I interferon signaling[13–16]. NDP52 supports replication of Chikungunya virus and measles virus

through the binding with non-structural viral proteins[17,18]. None of these NDP52 functions is involved in viral autophagy.

Hepatitis B virus (HBV) is an enveloped DNA virus with a partially double-stranded genome of 3.2 kb. Three envelope proteins S, M and L are produced by alternative translation initiations in a single open reading frame and harbor the common C-terminal domain (S). The M and L proteins contain an additional preS2 region at the N-terminus and the L protein has a further N-terminal extension called the preS1 region. Interaction of the L protein with HBV specific receptor sodium taurocholate cotransporting polypeptide (NTCP) via the preS1 region triggers viral endocytosis. Upon entry, the HBV virion delivers its

[1]University of Chinese Academy of Sciences, Chinese Academy of Sciences, 320 Yueyang Road, 200031 Shanghai, China. [2]Institut Pasteur, Université Paris Cité, 28 rue du Dr. Roux, 75015 Paris, France. [3]Shanghai Institute of Organic Chemistry, Chinese Academy of Sciences, 345 Lingling Road, 200032 Shanghai, China. [4]CIRI, Centre International de Recherche en Infectiologie, Univ Lyon, INSERM U1111, CNRS UMR5308, Université Claude Bernard Lyon 1, Ecole Normale Supérieure de Lyon, 69007 Lyon, France. [5]Department of Pathogen Biology, School of Basic Medical Sciences, Tianjin Medical University, 300070 Tianjin, China. ✉e-mail: yu.wei@pasteur.fr

genome into hepatocyte nucleus where it is converted into covalently closed circular DNA (cccDNA). cccDNA serves as template for viral transcription. In the cytoplasm, pregenomic RNA (pgRNA) is packaged with viral polymerase into the capsid in which the two strands of viral DNA are synthesized. The nucleocapsids bind to newly synthesized envelope proteins for secretion from the cell.

In HBV infection, both pro- and anti-viral effects of autophagy have been observed[19–22]. HBV infection activates autophagy and autophagic membranes could participate in viral nucleocapsid assembly, envelopment and viral release[23–25]. Silencing autophagy genes enhances HBV replication, suggesting autophagic degradation of viral products[26,27]. Viral protein HBx could dysregulate autophagy pathway by interacting with autophagy proteins[19,28–30].

Despite these findings, the functions of autophagy receptors in HBV infection have not yet been explored. In this work, we identify NDP52 as a sensor for HBV infection and demonstrate a specific mechanism by which NDP52 associates with Rab9 to control HBV infection.

## Results

### NDP52 recognizes the preS2 region of envelope proteins

In search for host factors implicated in HBV life cycle, we screened a human spleen cDNA library by a yeast two hybrid screening using viral open reading frames as bait. We identified NDP52 as an interactor of the preS2 region of envelope proteins (Fig. 1a). Glutathione S-transferase (GST) pulldown assays with GST fusion of NDP52 and Flag-tagged viral envelope proteins confirmed the interaction of NDP52 with L and M, but not S (Fig. 1b). When either individually transfected in Huh7 cells or expressed in HepAD38 cells replicating HBV, L and M, but not S, co-immunoprecipitated with endogenous NDP52 (Fig. 1c). Immunofluorescence assays in HepAD38 cells revealed a clear colocalization between NDP52 and M/L both in the absence and the presence of doxycycline (Dox) (Fig. 1d), indicating that the interaction of NDP52 with M/L is independent of viral replication. To assess the frequency of the interaction of L and M with NDP52, we expressed green fluorescence protein (GFP)-labeled L and M in Huh7 cells and performed immunofluorescence. GFP-labeled S was used as negative control. NDP52 colocalized with L and M with similar frequency (Fig. 1e, f).

HBV envelope proteins can localize in different organelles in the cell during viral replication. To specify the sub-compartments where M/L interact with NDP52, we performed immunofluorescence with ER-Tracker, anti-TGN46 or anti-CD63 antibodies for staining the endoplasmic reticulum (ER), Golgi apparatus and multivesicular body (MVB) respectively, to examine the locality of the interaction of viral envelope proteins with NDP52. The colocalization signals were detected in ER, Golgi and MVB (Fig. 1g). Therefore, NDP52 specifically recognizes HBV envelope proteins through interaction with the preS2 region in various organelles.

### NDP52 restricts HBV infection

We generated *NDP52*-knockout cell line in HepG2 cells stably expressing the viral receptor NTCP (NDP52[HepG2-NTCPKO]) (Fig. 2a) and confirmed that knockout of NDP52 does not impact on cell growth (Supplementary Fig. 1a). We infected NDP52[HepG2-NTCPKO] and NDP52[HepG2-NTCPWT] cells with HBV and examined viral replication. Analysis by Southern blot showed that NDP52 deletion increased cell susceptibility to HBV (Fig. 2b and Supplementary Fig. 1b), suggesting an inhibitory function of NDP52 in HBV infection. Complementation of NDP52[HepG2-NTCPKO] cells with NDP52 decreased the levels of viral replication (Fig. 2a, b), ruling out potential off-target effects of gene editing. Consistently, NDP52 overexpression in HepG2-NTCP cells was able to drastically suppress the virus (Fig. 2a, b). Analysis of viral DNA from cells and secreted virions by qPCR and secreted HBeAg, an active viral replication marker, by ELISA, further demonstrated the inhibitory effects of NDP52 on viral

replication (Fig. 2c–e). Analysis of viral DNA from *NDP52*-knockdown HepAD38 cells by Southern blot and qPCR confirmed the suppressor function of NDP52 on HBV (Supplementary Fig. 2a–d). Furthermore, we generated *NDP52*-knockdown Huh7 cells and transfected an overlength HBV genomic construct competent for viral replication. Analysis by qPCR showed significantly increase of viral DNA in *NDP52*-depleted cells (Supplementary Fig. 2e–g). These results indicate that NDP52 inhibits HBV replication.

### NDP52 depletion increases virus internalization

To investigate whether NDP52 is involved in the early steps of HBV infection, we performed HBV attachment and internalization assays in NDP52[HepG2-NTCPKO] and NDP52[HepG2-NTCPWT] cells. Cells were infected with HBV and collected 2 h post-infection for attachment assay and 16 h post-infection for internalization assay. No significant difference in viral attachment was observed between NDP52[HepG2-NTCPKO] and NDP52[HepG2-NTCPWT] cells (Supplementary Fig. 2h). In contrast, the level of HBV DNA internalized into cells was significantly enhanced by knock-out of NDP52 (Fig. 2f, left panel). Subsequently, the level of cccDNA was significantly increased in infected NDP52[HepG2-NTCPKO] cells (Fig. 2f, middle panel). Because cccDNA is the template of viral transcription, consequently, expression of pgRNA was at significantly higher level in NDP52[HepG2-NTCPKO] cells than NDP52[HepG2-NTCPWT] cells (Fig. 2f, right panel). Taken together, these results clearly indicate that NDP52 inhibits both HBV internalization and replication.

### Interaction of NDP52 with the preS2 region is required to restrict HBV

To identify the region of NDP52 responsible for the interaction with the preS2 region, we used M in coimmunoprecipitation assay with NDP52 deletion mutants (Fig. 3a). As shown in Fig. 3b, the interaction domain with the M protein was mapped in the galectin 8 interacting region (GIR)[31,32]. Deletion of GIR region in NDP52 (NDP52ΔGIR) drastically reduced the colocalization signals of NDP52 with M (see Fig. 3h, i) To test whether the GIR region was important for the activity of NDP52 on HBV, we introduced NDP52ΔGIR into NDP52[HepG2-NTCPKO] cells and assessed its effect on HBV. Despite similar levels of expression as wild-type NDP52, the ΔGIR mutant barely had effects on viral replication (Fig. 3c–f).

To determine the interaction domains in preS2, we constructed a mutant abolishing the two glycosylation sites in the preS2 region (mGly) and a series of M deletion mutants (Fig. 3g). GST pulldown assay showed that mutations in the preS2 glycosylation sites have no impact on the interaction with NDP52 and the region encompassing residues 13-26 is critical for the interaction with NDP52 (Fig. 3g). These results were further confirmed by immunofluorescence assays in which MΔ19 or MΔ26 mutants display significantly reduced colocalization with NDP52 (Fig. 3h, j).

The preS2 region is frequently found deleted in chronic hepatitis B patients[33]. To test the effect of NDP52 on preS2 mutants, we transfected pPreS2Δ19-26 or pPreS2WT with pHBVcore(-) (see Methods) in NDP52[HepG2-NTCPKO] and NDP52[HepG2-NTCPWT] cells and compared their replication capacity. In NDP52 wild-type control cells, the preS2Δ19-26 mutant showed significant higher levels of replication than preS2WT, whereas in NDP52 KO cells, no significant difference in replication was observed between preS2Δ19-26 and preS2WT (Fig. 3k). Taken together, these data indicate that the interaction with the preS2 region is required for NDP52 to exert its negative regulation on HBV.

NDP52 uses GIR domain to recognize galectin 8-bound *Salmonella* to trigger bacterial autophagy[31]. To examine the role of galectin 8 in HBV infection, we knocked down galectin 8 in HepG2-NTCP cells and found that depletion of galectin 8 had no effect on HBV replication (Supplementary Fig. 3a–d), indicating that galectin 8 is not involved in NDP52-mediated HBV inhibition.

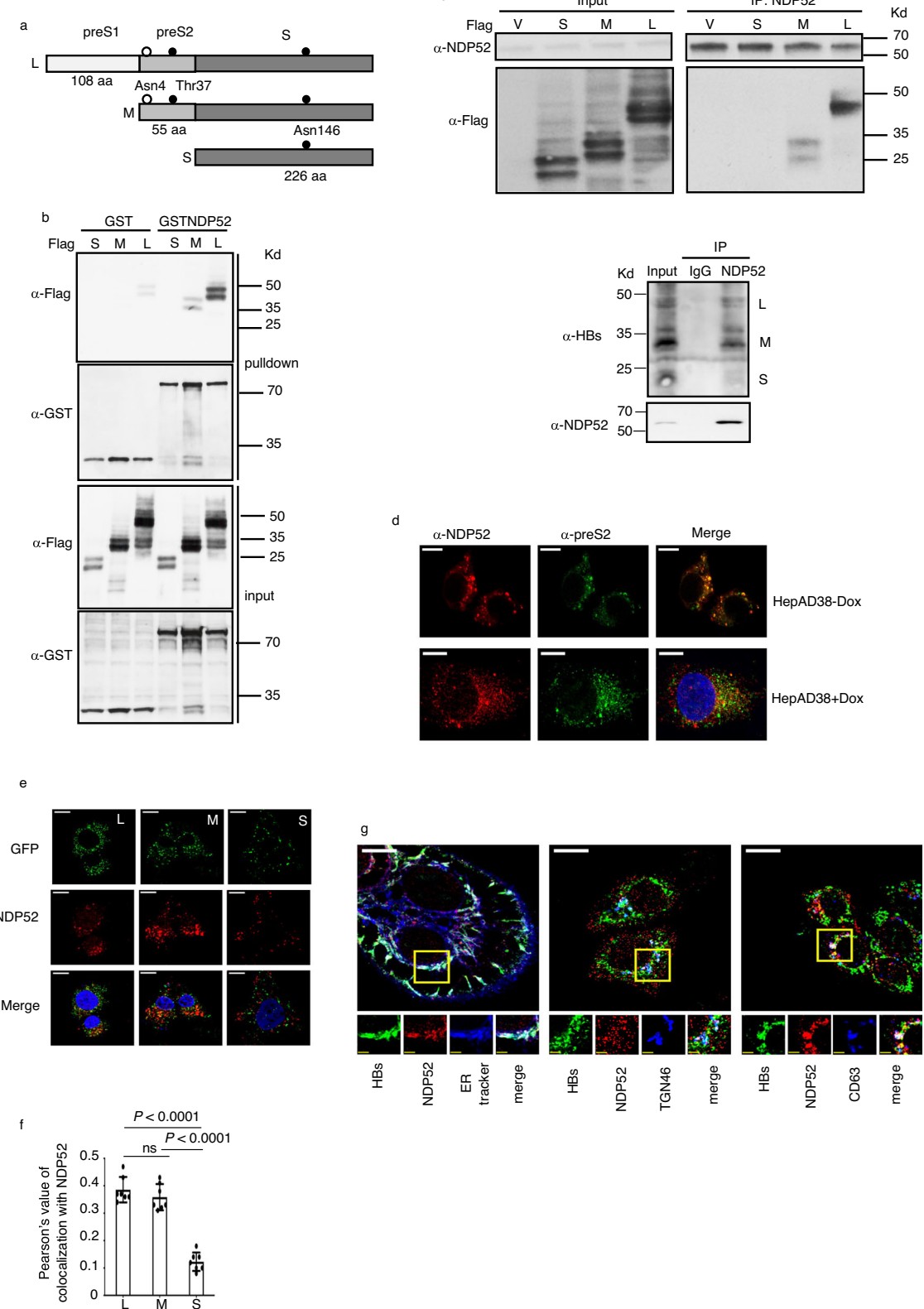

## NDP52 targets viral envelopes for lysosomal degradation

Immunofluorescence assays in HepAD38 cells showed a colocalization of viral envelope proteins (HBs), NDP52 and the lysosomal marker Lamp2 (Fig. 4a, upper panel). Western blot analysis showed envelope proteins in the lysosome (Fig. 4a, lower panel). Importantly, knockdown of NDP52 significantly decreases the colocalization of viral envelope proteins and Lamp2 (Fig. 4b). Following treatment with bafilomycin A1 to inhibit lysosome activity, the accumulation of envelope proteins, as well as autophagy substrate NDP52 was detected (Fig. 4c). Remarkably, depletion of NDP52 leads to substantial stabilization of envelope proteins and their pronounced insensitivity to the bafilomycin A1 treatment (Fig. 4c), suggesting that depletion of NDP52 specifically impairs the recruitment of envelope proteins to the lysosome. We further investigated this role of NDP52 by examining the

**Fig. 1 | NDP52 interacts with the preS2 region of HBV envelope proteins.**
**a** Schematic presentation of HBV three envelope proteins. The preS1, preS2 and S regions are indicated. Glycosylation within S and preS2 is indicated. **b** GST pull-down assays using cell extract of HEK293T cells transfected with plasmids encoding Flag-tagged S, M or L and GST or GST fusion of NDP52. Coprecipitated proteins were detected with indicated antibodies. **c** Coimmunoprecipitation assays with anti-NDP52 antibody in either Huh7 cells transfected with Flag-tagged vector (V), S, M or L (upper panel), or HepAD38 cells without doxycycline (Dox) (lower panel). Immunoprecipitates were detected by indicated antibodies.
**d** Immunofluorescence with anti-NDP52 and anti-preS2 antibodies shows colocalization of NDP52 with M and L in HepAD38 cells without Dox and with Dox. The

scale bar is 10 μm for full cell images. **e** Representative fluorescence micrographs of Huh7 cells transfected with plasmids coding for GFP fusion of L, M or S proteins and immunostained for NDP52. The scale bar is 10 μm for full cell images. **f** Pearson's correlation coefficients for colocalizations in **e** ($n = 7$ biological replicates). ns non significance. **g** Representative fluorescence micrographs of HepAD38 cells replicating HBV stained with anti-NDP52 and anti-HBs antibodies and reagents labeling ER (ER-Tracker), Golgi (anti-TGN46 antibody) and multivesicular body (anti-CD63). The scale bar is 10 μm for full cell images, 2.5 μm for zoomed images. Data are means ± SD. Statistical significance in **f** is determined by a two-sided unpaired t-test. Source data for **b**, **c** and **f** are provided as a Source Data file.

levels of viral envelope proteins in whole cell extract or in the lysosome. In whole cell extract, knockdown of NDP52 in HepAD38 cells resulted in stabilization and accumulation of envelope proteins (Fig. 4d). Notably, its levels were dramatic decreased in the lysosome in *NDP52*-depleted cells (Fig. 4d). Taken together, these data demonstrate an essential role for NDP52 in the distribution of envelope proteins in the lysosome.

We then assessed viral replication under the condition of the block in the lysosomal degradation. NDP52[HepG2-NTCPKO] and NDP52[HepG2-NTCPWT] cells were infected with HBV and then treated with ammonium chloride (NH₄Cl), a weak base that rapidly increases lysosomal pH. Southern blot analysis showed that $NH_4Cl$ treatment results in the increase of viral replicative DNA species (Fig. 4e). Notably, NDP52 deficiency leads to relative resistance to the $NH_4Cl$ treatment (Fig. 4e, compare line 3 versus line 1 with line 4 versus line 2). These observations were further validated by quantification of intracellular viral DNA (Fig. 4f). These findings indicate that NDP52 is essential for HBV clearance by lysosomal degradation.

### NDP52-mediated viral degradation is independent of ATG5

As NDP52 is an autophagy receptor, we investigated the relationship between the actions of NDP52 on HBV and autophagy protein machinery. Previous study showed that mutated NDP52[Y97A] or NDP52[A119Q] alleles abolish the formation of trimeric FIP200-NDP52-NAP1/SINTBAD complex that is required for NDP52-mediated bacterial autophagy[3]. We expressed wild-type NDP52, NDP52[Y97A] and NDP52[A119Q] in NDP52[HepG2-NTCPKO] cells and found that NDP52[Y97A] or NDP52[A119Q] can efficiently inhibit HBV replication as wild-type NDP52 (Supplementary Fig. 4a–d), indicating that trimeric complex formation with FIP200 and NAP1/SINTBAD is not required for NDP52 to mediate HBV clearance.

ATG5 is an essential component of LC3-lipidation process, which is required for the formation of autophagosomes in canonical autophagy. However, increasing body of evidence has shown that autophagosome biogenesis can be achieved through alternative pathways[34–36]. To determine the role of ATG5 in the actions of NDP52 on HBV, we generated ATG5 knockout HepG2-NTCP cell lines (ATG5[HepG2-NTCPKO]) and overexpressed NDP52 in ATG5[HepG2-NTCPKO] (Supplementary Fig. 5a, b). Deficiency of ATG5 resulted in diminished HBV replication (Supplementary Fig. 5c), consistent with a previous report which showed that canonical autophagy pathway supports viral replication[20]. Notably, NDP52 can significantly restrict viral replication in ATG5[HepG2-NTCPKO] cells (Supplementary Fig. 5d, e), indicating that LC3 lipidation is not required for NDP52-mediated HBV suppression. Furthermore, we did not observe substantial colocalization of viral envelope proteins, NDP52 and LC3 in HepAD38 cells (Supplementary Fig. 5f).

### Rab9 is crucial for NDP52-mediated HBV inhibition

Previous study reported that an ATG5-independent alternative autophagy pathway is regulated by Rab9, which is initiated with membranes derived from *trans*-Golgi and late endosomes[34]. We observed a clear colocalization of HBs, NDP52 and Rab9, as well as

HBs, Rab9 and Lamp2 in HepAD38 cells (Fig. 5a). Depletion of NDP52 had no effect on the colocalization of HBs and Rab9 (Fig. 5b). We found that endogenous Rab9 and NDP52 coimmunoprecipitated in HepAD38 cells (Fig. 5c). As viral envelope proteins are constitutively expressed in HepAD38 cells and independent of doxycycline control, we used Huh7 cells to assess the role of viral envelopes in the interaction of NDP52 and Rab9. Interestingly, Rab9 interacted with M and only immunoprecipitated NDP52 when M was present (Fig. 5d, e).

It was shown that the cytotoxic stressor etoposide and starvation induce both canonical and alternative autophagy, whereas rapamycin is able to only induce Atg5-dependent autophagy[34]. We treated HepAD38 cells with etoposide, starvation and rapamycin in combination with bafilomycin A1 and analyzed the levels of viral envelope proteins. In whole cells, the treatments resulted in the accumulation of NDP52, Rab9 and Lamp2 (Fig. 5f), which are cellular substrates of autophagy. No accumulation of viral envelope proteins was detected following combination treatments compared to bafilomycin A1 treatment (Fig. 5f), which could be explained by more secretion of viral particles outside cells. However, in the lysosome, the levels of viral envelope proteins were largely augmented by the treatments, indicating that stimulation of autophagy causes envelope protein lysosomal targeting (Fig. 5f). Interestingly, while NDP52, Rab9 and Lamp2 were accumulated in the lysosome at similar levels among etoposide, starvation and rapamycin treatments, the accumulation of envelope proteins was more pronounced upon etoposide and starvation treatments than that of rapamycin (Fig. 5f), suggesting that Rab9-dependent pathway plays an important role in lysosomal targeting of viral envelopes. Knockdown of Rab9 increased envelope proteins in whole cell lysis, but decreased them in the lysosome (Fig. 5g). Of note, Rab9 knockdown also decreased NDP52 in the lysosome, but had no effect on Lamp2 (Fig. 5g). These findings demonstrate that Rab9 is a key component in lysosomal targeting and degradation of viral envelopes.

To dissect the mechanisms of NDP52 and Rab9 cooperation, we knocked down Rab9 in HepG2-NTCP cell lines (Rab9[HepG2-NTCPKD]) (Fig. 5h). Depletion of Rab9 resulted in more active HBV replication with the increase of viral DNA species in cells and HBeAg in the medium (Fig. 5i–k), indicating that downregulation of Rab9-dependent lysosomal degradation pathway promotes HBV replication. We overexpressed NDP52 in *Rab9*-knockdown cells (see Fig. 5h), and assessed the ability of NDP52 to regulate HBV. Knockdown of Rab9 abrogated the ability of NDP52 to inhibit viral replication (Fig. 5l–n). These data demonstrate that NDP52 restricts HBV infection via Rab9-dependent lysosomal degradation pathway.

### NDP52 suppresses HBV in vivo

We investigated the effects of NDP52 on HBV in vivo in a mouse model for chronic hepatitis B. HBV infection was established by transduction of mice with recombinant adeno-associated virus HBV (AAVHBV)[37]. We used adenovirus vector to express human NDP52 (AdNDP52) in the liver. $5 \times 10^{10}$ AdNDP52 were inoculated by tail vein injection into mice 4 weeks after AAVHBV transduction (week 0). Beginning at 2 weeks,

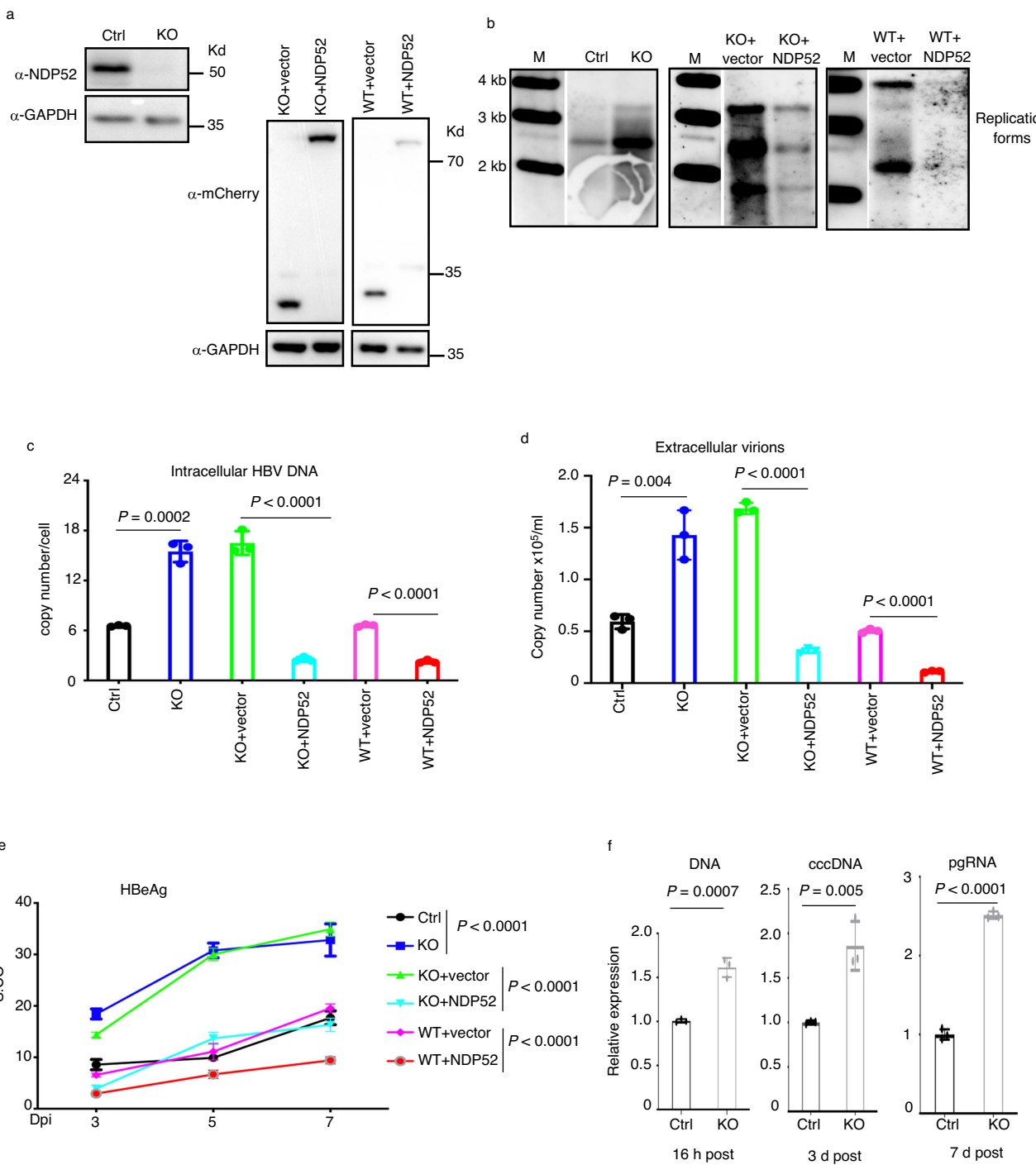

**Fig. 2 | NDP52 restricts HBV infection. a** Immunoblot analysis of indicated proteins from cellular extract of CRISPR control NDP52[HepG2-NTCPWT] (Ctrl), NDP52[HepG2- NTCPKO] (KO), NDP52[HepG2-NTCPKO] restored with mCherry tag vector or mCherry-tagged NDP52 (KO + vector, KO + NDP52), or NDP52[HepG2-NTCPWT] overexpressing mCherry tag vector or mCherry-tagged NDP52 (WT + vector, WT + NDP52) cells. **b** Indicated cells were infected with HBV. Cytoplasmic core DNA was purified fourteen days post infection and detected by Southern blot analysis using an HBV DNA probe. M: DNA marker. Depending on infection, variations in detected viral replication species were observed. Therefore, ensemble replicative species are designated as replication forms. Quantitative PCR of HBV DNA from infected cells (**c**) and extracellular virions (**d**) seven days post infection ($n = 3$ biological replicates). **e** Medium levels of viral HBeAg released from infected cells ($n = 3$ biological replicates). S:CO signal-to-cutoff ratio, Dpi day post infection. **f** NDP52[HepG2-NTCPWT] and NDP52[HepG2-NTCPKO] cells were infected with HBV. Cells were collected at indicated time. Viral DNA, cccDNA and pregenomic RNA (pgRNA) were analyzed by qPCR. Expression in Ctrl cells was arbitrarily set as 1 ($n = 3$ biological replicates). Data are means ± SD. Statistical significance in **c**–**f** is determined by a two-sided unpaired t-test. Source data for **a**–**f** are provided as a Source Data file.

there was a significant decrease in HBsAg and HBeAg in mice receiving AdNDP52 as compared to mice injected with empty adenovirus (AdCtrl) (Fig. 6a, b). Although there was a significant change in serum alanine aminotransferase (ALT) following administration of AdNDP52 in both AAVHBV-transduced mice and naïve mice, the levels of ALT in AAVHBV-transduced mice were much higher than those in naïve mice (Fig. 6c and Supplementary Fig. 6a). Immune cell infiltration, however, was barely detectable 3 weeks post Ad injection (Supplementary

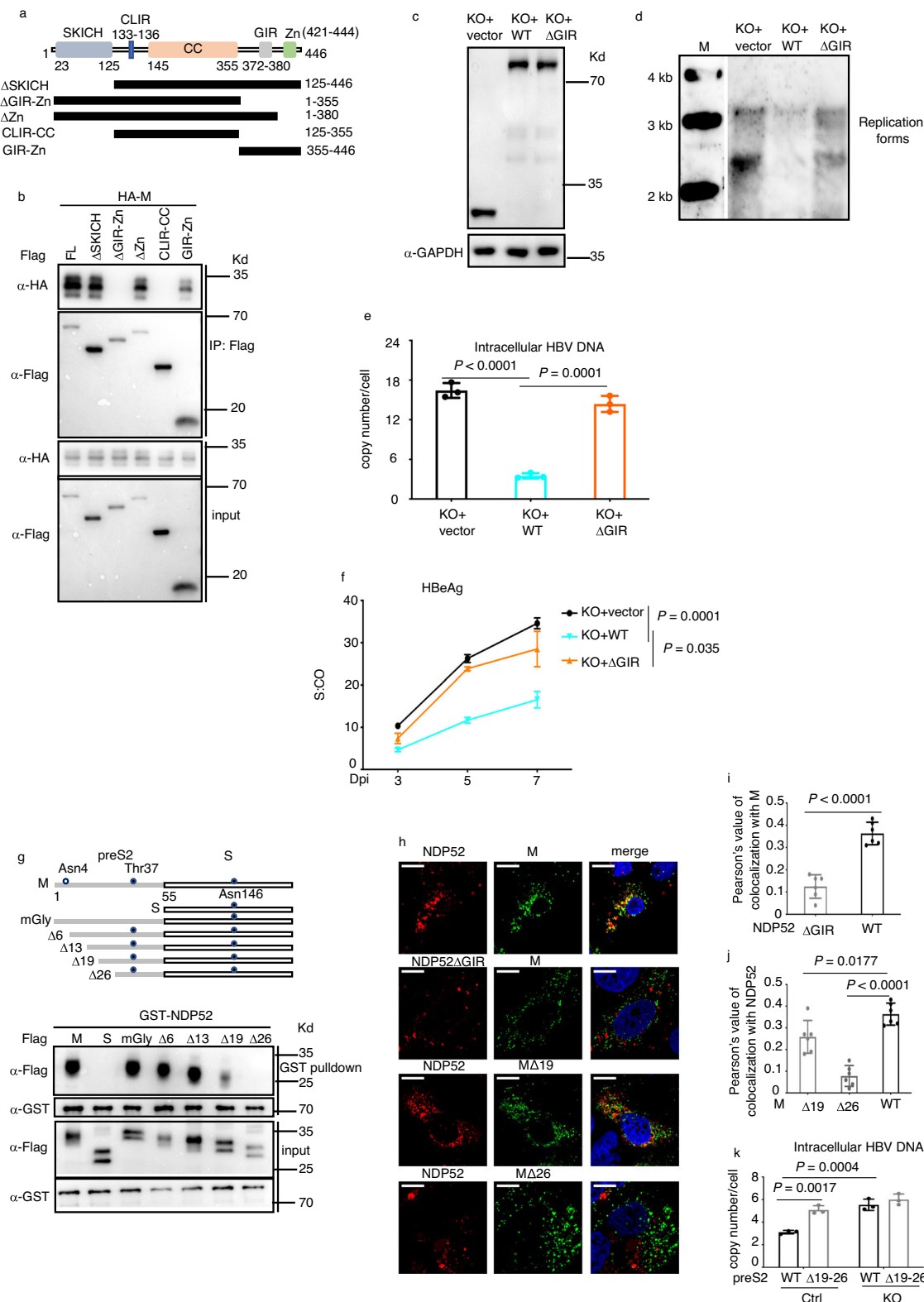

Fig. 6b). Intrahepatic HBV DNA, pgRNA, envelope proteins and core protein were examined by qPCR or immunohistochemistry at 3 weeks post Ad injection. Significant reduction in HBV DNA and pgRNA was observed in the liver of mice injected with AdNDP52, which also showed less core-expressing hepatocytes and diminished levels of HBsAg (Fig. 6d, e). Examination of CD8[+] T cells in liver draining lymph nodes and spleens showed that significant CD8[+] T cell activation was observed only in the spleen of mice receiving AdNDP52 at 2 weeks

(Supplementary Fig. 6c), suggesting moderate induction of T-cell response. These data suggest that NDP52 may be important for HBV clearance in vivo.

## Discussion

Hepatitis B virus can evade host surveillance and develop chronic infection. In this work, we demonstrate NDP52 as a host sensor for viral proteins and define the role of NDP52 against HBV infection.

**Fig. 3 | NDP52 interaction with preS2 is required for inhibition of HBV. a** A schematic map of NDP52 showing the domain organization and the deletion constructs used to identify the interaction domain with the preS2 region. SKICH skeletal muscle and kidney enriched inositol phosphatase carboxyl homology domain, CLIR LC3C interacting region, CC coiled-coil, GIR galectin 8 interacting region, Zn zinc finger. **b** Coimmunoprecipitation assays using cell extract of HEK293T cells transfected with plasmids encoding Flag-tagged NDP52 full length (FL) or truncations and HA-tagged M envelope protein. Flag-coprecipitated proteins were detected with indicated antibodies. **c** Immunoblot analysis of indicated proteins from cellular extract of NDP52[HepG2-NTCPKO] cells restored with mCherry tag vector, mCherry-tagged wild-type NDP52 (WT) or mCherry-tagged NDP52 lacking GIR domain (ΔGIR). **d** Indicated cells were infected with HBV. Core DNA was purified fourteen days post infection and detected by Southern blot analysis using an HBV DNA probe. M: DNA marker. **e** Quantitative PCR of HBV DNA from infected cells seven days post infection (n = 3 biological replicates). **f** Medium levels of HBeAg

from infected cells. S:CO signal-to-cutoff ratio, Dpi day post infection (n = 3 biological replicates). **g** GST pull-down assays using cell extract of HEK293T cells transfected with plasmids encoding Flag-tagged S and M constructs and GST fusion of NDP52. GST pulldown proteins were detected with indicated antibodies. In schematic presentation of S and M constructs, glycosylation sites are presented. mGly: mutant with the two glycosylation sites Asn4 and Thr37 at the preS2 region changed to Ala4 and Ala37. **h** Representative fluorescence micrographs of Huh7 cells transfected with plasmids coding for mCherry-NDP52 and Flag-M and immunostained with anti-Flag antibody. The scale bar is 10 μm for full cell images. **i, j** Pearson's correlation coefficients for colocalizations in **h** (n = 6 biological replicates). **k** Quantitative PCR of HBV DNA from NDP52[HepG2-NTCPWT] (Ctrl) and NDP52[HepG2-NTCPKO] (KO) cells transfected with pPres2Δ19-26 or pPreS2WT with pHBVcore(-) four days post transfection (n = 3 biological replicates). Data are means ± SD. Statistical significance in **e, f, i–k** is determined by a two-sided unpaired t-test. Source data for **b–g, i–k** are provided as a Source Data file.

NDP52 recognizes the preS2 region of viral envelope proteins, which is common to M and L. The L protein displays two transmembrane topologies with the preS region either protruding into the ER lumen or exposed on the cytosolic side of the ER membrane, whereas M exhibits one topology with the preS2 region in the ER lumen[38–41]. The preS on the cytosolic side is required for binding to viral nucleocapsids, whereas the preS in the ER lumen is displayed on the surface of mature virions. By immunofluorescence, we detected the colocalization of NDP52 with both M and L in a similar frequency. During viral life cycle, envelope proteins can localize in different organelles in the cell. Colocalization signals with NDP52 were found in ER, Golgi and MVB. In addition, we show that NDP52 affects HBV 16 h post infection, suggesting that NDP52 may recognize the preS2 region at the early step of viral life cycle. The mechanism by which NDP52 binds to the preS2 region of M and L is not yet known. The topology of viral envelope proteins in each organelle, its dynamics during viral replication and the relationship between viral envelope proteins and host factors are the questions that remain to be answered. Given the highly dynamic nature of the Dane particle and subviral particles, it is safe to assume that there are still unknown aspects in the topology of HBV envelope proteins in the course of viral life cycle, which will be progressively unveiled with the development of new technology and research tools. We speculate that in the delivery process of viral envelope proteins during viral replication, the preS2 region in L and M may be exposed to NDP52-containing cell compartments to allow the interaction with it. Further studies are needed to better elucidate the mechanisms of the recognition of the preS2 region by NDP52, which would shed important insights on the evolution of L and M topology in each step of viral life cycle. Although direct evidence of NDP52 targeting of enveloped viruses into the lysosome is lacking, the observation of the increase of core viral DNA in *NDP52⁻ᐟ⁻* cells provides strong support that NDP52 mediates degradation of enveloped viruses. Two major lines of evidence suggest that NDP52 recognition of envelope proteins occurs in both viral entry and post-entry life cycle. First, depletion of NDP52 in HepG2-NTCP cells significantly increased the level of HBV DNA internalized into cells. Second, the activity of HBV replication in post-entry life cycle was enhanced in *NDP52*-depleted HepAD38 and Huh7 cells, which bypass the entry step to replicate the virus, suggesting that NDP52 suppresses newly assembled viruses.

Our studies suggest a mechanism implicating lysosomal clearance by which NDP52 protects hepatocytes against HBV infection. Using genetic, biochemical and virological analyses, we show that lysosome inhibition increases viral replication and viral envelope protein accumulation. We demonstrate that depletion of NDP52 significantly decreases the targeting of envelope proteins, including enveloped virions, into the lysosome, as in *NDP52⁻ᐟ⁻* cells both envelope proteins and viral DNA were increased. The failure of NDP52ΔGIR mutant, which does not bind to preS2, to rescue the function in inhibiting HBV replication is consistent with the substrate-targeting role of NDP52 in

selective autophagy. Currently, it is not known whether NDP52-targeted HBV envelopes are delivered into the lysosome through double-membrane autophagosomes. The observations that autophagy stimuli such as starvation, etoposide and rapamycin induce accumulation of M and L in the lysosome suggest that selective autophagy pathway may be involved in HBV degradation. Since its identification as an autophagy receptor targeting *Salmonella* to degradation[10], the research on NDP52 has yielded a range of data attesting to its role as antibacterial effector of the innate immunity[7,11,12]. However, no specific protein in bacteria was identified as target of NDP52 in bacterial autophagy. Although NDP52 has been shown to interact with viral proteins of diverse viruses[14–18,42], none of these viral proteins was targeted by NDP52 for degradation. Here we provide evidence that NDP52 specifically targets HBV envelope proteins into the lysosome, which constitutes the first example of NDP52 targeting of a specific microbial protein for lysosomal degradation.

Our findings establish the involvement of both ATG5-dependent canonical autophagy and an ATG5-independent pathway in the regulation of HBV infection. Our data with ATG5 depletion, which decreases virus production, confirm that ATG5-dependent canonical autophagy is important for efficient viral replication[19,20]. Our studies demonstrate a previously unappreciated role for NDP52-mediated HBV lysosomal degradation in viral clearance, which does not rely on ATG5. In a previous study, it was shown that Rab9 promotes autophagosome biogenesis with membranes derived from the trans-Golgi and late endosomes[34]. This ATG5-independent Rab9-dependent alternative autophagy can be triggered in response to certain cellular stress and during erythrocyte differentiation in vivo[34]. Our findings demonstrate that Rab9 depletion increases the susceptibility to HBV and mitigates NDP52-mediated suppression on the virus, indicating that the ability of NDP52 to trigger HBV degradation is influenced by Rab9. We found that NDP52, Rab9 and M proteins form a complex and that NDP52 interacts with Rab9 in M-dependent manner. The susceptible phenotype in *NDP52*-overexpressed *Rab9*-knockdown cells indicates that virus-induced Rab9 recruitment is crucial for NDP52 to exert its inhibitory function on the virus. Previous study reported that Rab9-dependent alternative autophagy is regulated by ULK complex and beclin 1[34]. Our observations that NDP52 mutants incapable of forming trimeric complex with FIP200 and NAP1/SINTBAD still have the ability to inhibit HBV suggest that NDP52 interaction with ULK complex is not required for HBV degradation. Although it is not yet known the precise signaling events that connect the detection of HBV by NDP52 to Rab9-dependent pathway, our findings suggest that NDP52 may promote viral delivery through interaction with the Rab9-associated nascent membrane derived from trans-Golgi and late endosomes. Further studies are needed to determine how NDP52-Rab9-viral envelopes interaction takes place, whether autophagosomes are formed, whether NDP52/Rab9 pathway targets other viruses to lysosomal degradation. Of interest, our data suggest that, like canonical autophagy,

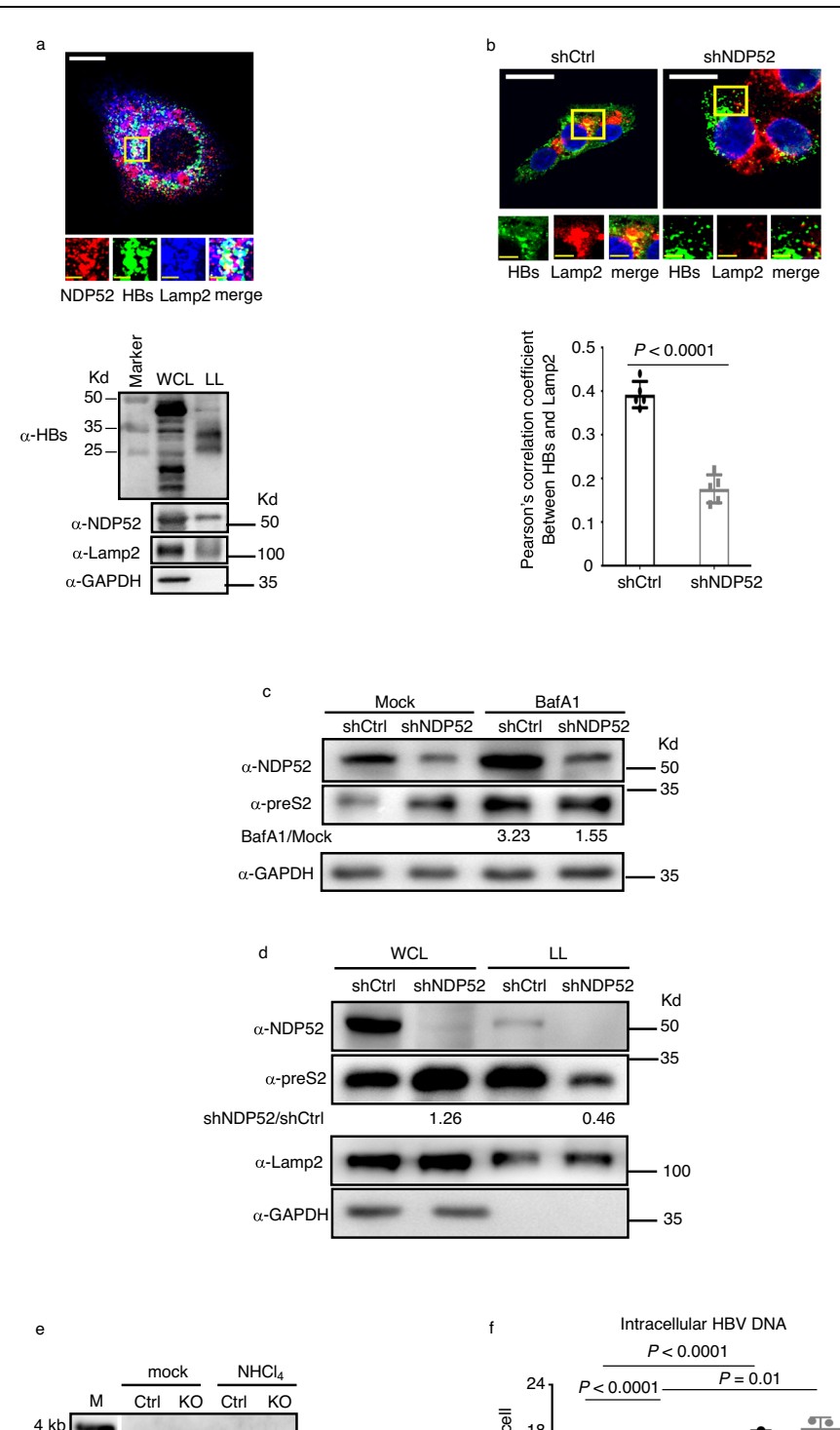

Rab9-dependent lysosomal degradation pathway can be also selective, which might involve similar set of autophagy receptors.

In a previous report, Rab9 was not identified as HBV regulator in siRNA screening in HepG2.2.15 cells[43]. HepG2.2.15 is a stable cell line containing four copies of HBV genome in which the virus replicates actively. The efficiency of Rab9 silencing by siRNA transfection in HepG2.2.15 was not known. In the current study, we knocked down Rab9 expression with shRNA in HepG2-NTCP cells and infected Rab9$^{HepG2-NTCPKD}$ cells with HBV. Rab9 expression was efficiently reduced by Rab9-specific shRNA (Fig. 5g). We showed that knockdown of Rab9 increased HBV replication. The discrepancy of the two studies may be attributed to the difference in the sensitivity of the methods.

**Fig. 4 | NDP52 targets viral envelopes to lysosomal degradation.**
**a** Colocalization of NDP52, viral envelope proteins (HBs) and Lamp2 in HepAD38 cells (upper panel). The scale bar is 10 μm for full cell images, 2.5 μm for zoomed images. Immunoblot analysis of indicated proteins from either whole cell lysate (WCL) or lysosome lysate (LL) of HepAD38 cells in the presence of Dox (lower panel). The major band close to 35 kd was also detected by anti-preS2 antibody and is shown in the other figures of WB with lysosome extract. **b** Short hairpin RNAs targeting NDP52 (shNDP52) or non-targeting control (shCtrl) were expressed in HepAD38 cells. *NDP52*-knockdown HepAD38 or control cells were immunostained for viral envelope proteins (HBs) and Lamp2. The scale bar is 10 μm for full cell images, 2.5 μm for zoomed images. Pearson's correlation coefficients for colocalization are presented (*n* = 5 biological replicates). **c** Immunoblot analysis of indicated proteins from cellular extract of HepAD38 cells mock-treated or treated with

bafilomycin A1 (BafA1). The ratio of the envelope protein normalized to GAPDH in BafA1-treated cells versus mock-treated cells is presented. **d** Immunoblot analysis of indicated proteins from either whole cell lysate (WCL) or lysosome lysate (LL) of HepAD38 cells in the presence of Dox. The ratio of the envelope protein normalized to GAPDH or Lamp2 in shNDP52 cells versus shCtrl cells is presented. **e** NDP52$^{HepG2-NTCPWT}$ (Ctrl) and NDP52$^{HepG2-NTCPKO}$ (KO) cells were infected with HBV and treated with NHCl$_4$. Cytoplasmic core DNA was purified fourteen days post infection and detected by Southern blot analysis using an HBV DNA probe. M: DNA marker. **f** NDP52$^{HepG2-NTCPWT}$ and NDP52$^{HepG2-NTCPKO}$ cells were infected with HBV and treated with NHCl$_4$. Quantitative PCR of HBV DNA from infected cells seven days post infection (*n* = 3 biological replicates). Data are means ± SD. Statistical significance in **b** and **f** is determined by a two-sided unpaired t-test. Source data for **a**–**f** are provided as a Source Data file.

Increasing evidence has shown a critical role for galectin 8 in host defense against pathogens[31,44,45]. Although the domain of NDP52 interacting with preS2 is mapped to the GIR domain, depletion of galectin 8 has no effect on HBV replication, suggesting that galectin 8 is not involved in NDP52 recruitment to viral envelope proteins.

The preS2 domain is frequently involved in deletions and mutations leading to the emergence of HBV variants in patients[33]. These variants are associated with severe forms of acute and chronic liver diseases, which is strongly consistent with a defect in adaptive immune response due to the deletion of T- and B-cell epitopes[46–48]. Besides, it is plausible that the failure to properly recognize and clear these variants by NDP52 may contribute to the pathogenesis. This notion is supported by our in vitro study showing the increased replication capacity of the preS2 mutant that lacks NDP52-interacting domain. Conversely, NDP52-mediated HBV lysosomal destruction may constitute a selection pressure for the emergence of variants that escape NDP52 targeting. Our data with the enforced expression of NDP52 in the liver, which largely suppresses chronic HBV replication, provide evidence that NDP52 might orchestrate lysosomal degradation of the virus in vivo. Although AdNDP52 vector significantly induced ALT levels, the absence of immune cell infiltration and marginal T cell activation in the liver suggest that HBV clearance is linked to the function of NDP52.

Our study suggests that there are distinct autophagy receptor-mediated mechanisms for activating pathogen degradation in response to microbial infections. We speculate that future research on autophagy receptors in infection will uncover novel mechanisms by which these autonomous factors function as crucial mediators of host immunity.

## Methods

### Yeast two-hybrid screen
The yeast two-hybrid screen protocol was previously described[49]. Briefly, HBV ORFs were cloned in the vector pPC97-GW using the Gateway technology (Invitrogen). AH109 yeast cells were transformed with plasmids expressing viral proteins fused to the DNA-binding domain of GAL4 (GAL4-BD). In parallel, a human spleen cDNA library (Invitrogen) cloned in the pPC86 vector was transformed in Y187 yeast cells to express cellular proteins fused to the transactivation domain of GAL4 (GAL4-AD). Then, AH109 and Y187 cells were mated and plated on a synthetic medium supplemented with 3-amino-1,2,4-triazole (3-AT) and lacking histidine (-His). Plates were incubated for 6 days at 30 °C and yeast colonies were picked and rearrayed. Yeast colonies were lysed and cDNA inserts were amplified with flancking primers that hybridize into the pPC86 vector. PCR products were sequenced and cellular interactors were identified by BLAST analysis. A total of 13 positive yeast colonies corresponding to NDP52 were retrieved from the screen with preS2.

### Plasmids
Plasmids encoding GST fusion of human NDP52 was cloned into pDEST. Plasmids coding for 3xFlag-tagged L, M, S proteins and M

mutants were cloned into pCL-neo. Plasmids encoding Flag-tagged NDP52 mutants were cloned in pCR3, as described previously[7]. Plasmids encoding GFP-L, GFP-M, GFP-S and HA-tagged M were cloned into pCDH-CMV-puro. Plasmid GFP-Rab9 was obtained from Addgene (#12663)[50].

To test the effects of NDP52 on preS2 mutants, pPreS2Δ19-26, pPreS2WT and pHBVcore(-) were constructed based on pHBVawy1.3. pPreS2Δ19-26 was constructed with following primers: preS2 19-26 deletion forward 1 5′- GAGTGAGAGGCCTGTATTTCTCCAGTTCAGGA ACAGTAAA-3′ and reverse 1 5′- TTTACTGTTCCTGAACTGGAGAAATA-CAGGCCTCTCACTC-3′, forward 2 5′- TGCAAGATCCCAGAGTGAGATC CAGTTCAGGAACAGTAAA-3′ and reverse 2 5′- TTTACTGTTCCTGAAC TGGATCTCACTCTGGGATCTTGCA-3′. The overlapping viral polymerase ORF at the amino acid positions 308-315 is also deleted in pPreS2Δ19-26 and the arginine at the position 307 is changed to isoleucine. Although the 308-315 deletion occurs in the spacer domain of the polymerase, full length polymerase was provided in trans to ensure efficient viral replication. To construct the control plasmid, a stop codon was introduced at the amino acid position 708 in the polymerase ORF in pHBVawy1.3 to abrogate expression of the polymerase. This plasmid was named pPreS2WT. The following primers were used for the construction of pPreS2WT: polymerase stop codon forward 5′-CGGCCAGGTCTGTGC**TAA**GTGTTTGCTGACGCA-3′ and reverse 5′-TGCGTCAGCAAACAC**TTA**GCACAGACCTGGCCG-3′. The stop codon is underlined. pHBVcore(-) was constructed as previously described[51]. The start codon of the core in pHBVawy1.3 was replaced by a stop codon so that pHBVcore(-) is replication-incompetent by itself, but can provide polymerase expression in trans. The following primers were used: core deletion forward 5′- TGC CTTGGGTGGCTTTGGGGC**TGA**-GACATCGACCCTTATAAAGAA-3′ and reverse 5′- TTC TTTATAAGGG TCGATGTC**TCA**GCCCCAAAGCCACCCAAGGCA-3′. The stop codon is underlined.

pCDH-based constructs were transformed and amplified in MAX Efficiency™ Stbl2™ competent cells (Thermo Fisher Scientific). The other constructs were transformed and amplified in the *Escherichia coli* DH5α competent cells (Thermo Fisher Scientific). All plasmids were verified by sequencing (Tsingke Biotechnology Co., Ltd.)

### Cells
HepG2 (ATCC #HB-8065) and HEK293T (ATCC#CRL-3216) cell line were from ATCC. HepAD38 cells were kindly provided by Dr. Christoph Seeger (Fox Chase Cancer Center, Philadelphia, USA). Huh7 cells, established by Sato, J., and Nakabayshi, H., in 1982, have been widely distributed in research laboratories woldwide since then.

The human hepatoma cell lines HepG2 and Huh7 as well as HEK293T cells were cultured in Dulbecco's Modified Eagle Medium (DMEM) (Biological Industries, 01-055-1A) supplemented with 10% fetal bovine serum (FBS) (YLESA, S211201T), 100 U ml⁻¹ penicillin and 100 μg ml⁻¹ streptomycin (Biological Industries, 03-031-1B). Media supplemented with heat-inactivated FBS were used for viral infection experiments. To maintain the cell line, HepG2-derived HepAD38 cells

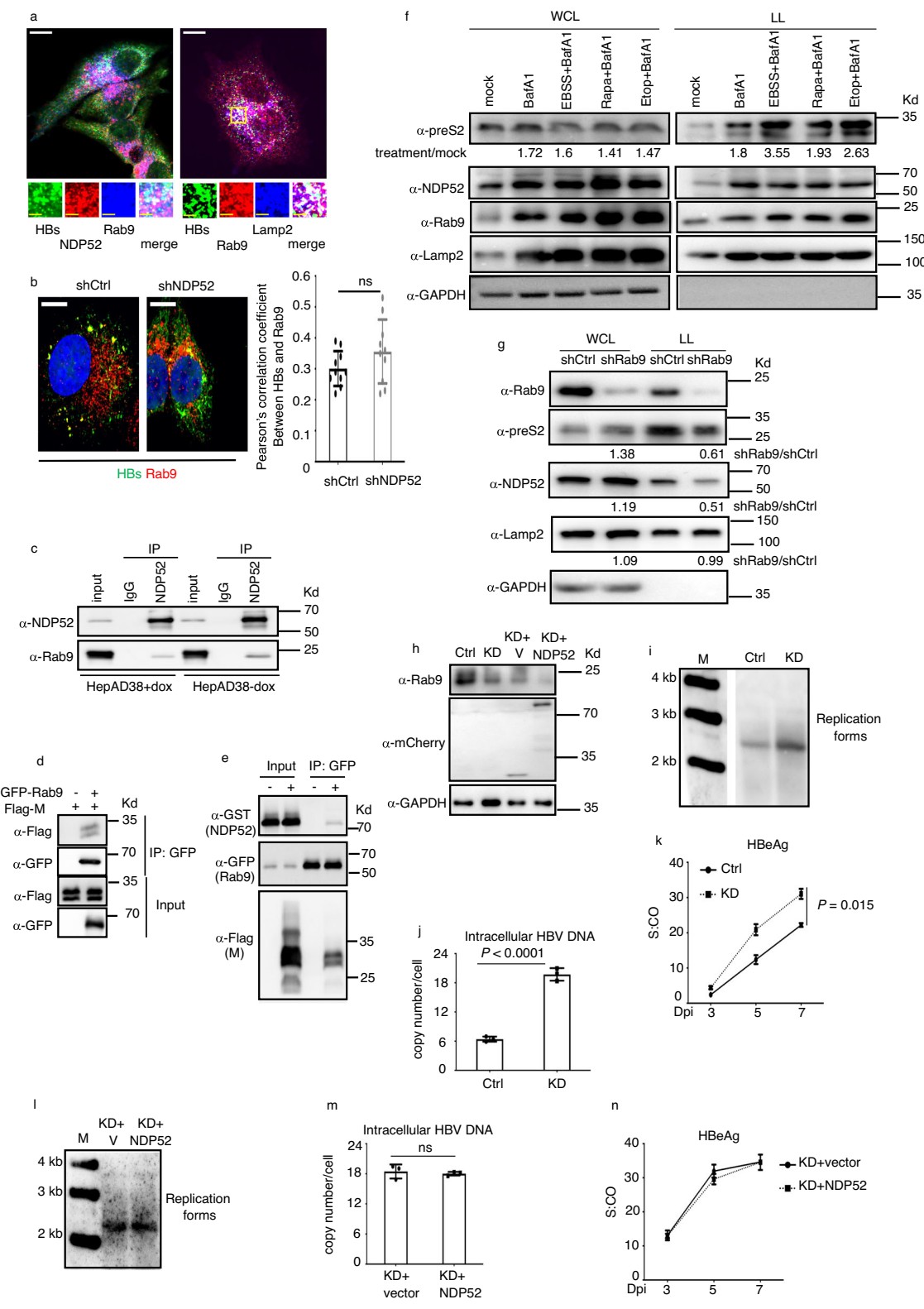

containing integrated HBV genome from which pregenomic RNA transcription is under the control of doxycycline were cultured in DMEM-F12 medium (Biological Industries, 01-170-1A) supplemented with 10% FBS, 100 units ml⁻¹ penicillin and 100 µg ml⁻¹ streptomycin, 400 µg ml⁻¹ G418 (Millipore, 345810), 1 µg ml⁻¹ doxycycline (Sigma Aldrich, D9891). Viral replication is induced when doxycycline is removed[52]. HepG2-NTCP cells were generated by infection of lentivirus expressing human NTCP[53] and clonal selection in medium containing

10 µg ml⁻¹ blasticidin (Invitrogen, R210-01). HepG2-NTCP *Atg5* and HepG2-NTCP *NDP52* CRISPR-knockout cell lines (ATG5$^{HepG2-NTCPKO}$, NDP52$^{HepG2-NTCPKO}$) were generated by CRISPR/Cas9-mediated genome editing using gRNA guides cloned into pLentiCRIPRv2 puro plasmid (Addgene #98290) for simultaneous gRNA and Cas9 expression[54]. HepG2-NTCP CRISPR control cells were generated using pLenti-CRIPRv2 puro empty vector. Cells were selected with 2 µg ml⁻¹ pur-omycin (Invivogen, A1113803) and single-cell colonies were obtained

**Fig. 5 | Rab9 is involved in NDP52-mediated viral degradation. a** Colocalization of HBs, NDP52 and Rab9 or HBs, Rab9 and Lamp2 in HepAD38 cells. The scale bar is 10 μm for full cell images, 2.5 μm for zoomed images. **b** NDP52 knockdown HepAD38 (shNDP52) or knockdown control cells (shCtrl) were immunostained for HBs and Rab9 ($n = 10$ biological replicates). The scale bar is 10 μm for full cell images. Pearson's correlation coefficients for colocalization are presented. ns: non significance. **c** Coimmunoprecipitation assays using HepAD38 cells. Rab9 is coprecipitated with NDP52. **d** Coimmunoprecipitation assays with anti-GFP antibody in Huh7 cells transfected with GFP-tagged Rab9 and Flag-tagged M. **e** Coimmunoprecipitation assays with anti-GFP antibody in Huh7 cells transfected with GFP-tagged Rab9, GST-tagged NDP52 and Flag-tagged M. **f** HepAD38 cells were treated with bafilomycin A1 (BafA1), Earl's Balanced Salts (EBSS), rapamycin (Rapa) or etoposide (Etop). Proteins from either whole cell lysate (WCL) or lysosome lysate (LL) were analyzed by immunoblot assay. The ratio of the envelope protein expression in WCL and LL normalized to GAPDH or Lamp2 respectively in reagent-treated cells versus mock-treated cells is presented. **g** Immunoblot analysis of indicated proteins from either WCL or LL of *Rab9*-knockdown (shRab9) or control (shCtrl) HepAD38 cells. The ratio of protein expression in shRab9 cells versus shCtrl

cells in WCL and LL normalized to GAPDH or Lamp2 respectively is presented below each graph. **h** Immunoblot analysis of indicated proteins from cellular extract of Rab9 knockdown control Rab9$^{HepG2-NTCPWT}$ (Ctrl), Rab9$^{HepG2-NTCPKD}$ (KD), Rab9$^{HepG2-NTCPKD}$ overexpressing mCherry-tagged NDP52 (KD + NDP52) or mCherry tag vector cells (KD + vector). **i** Rab9$^{HepG2-NTCPWT}$ (Ctrl), Rab9$^{HepG2-NTCPKD}$ (KD) cells were infected with HBV. Core DNA was purified fourteen days post infection and analyzed by Southern blot using an HBV DNA probe. M: DNA marker. **j** Quantitative PCR of HBV DNA from infected cells ($n = 3$ biological replicates). **k** Medium levels of HBeAg from infected cells ($n = 3$ biological replicates). S:CO signal-to-cutoff ratio, Dpi day post infection. **l** Rab9$^{HepG2-NTCPKD}$ overexpressing NDP52 (KD + NDP52) or control cells (KD + vector) cells were infected with HBV. Core DNA was purified and analyzed by Southern blot using an HBV DNA probe. **m** Quantitative PCR of HBV DNA from infected cells seven days post infection ($n = 3$ biological replicates). ns non significance. **n** Medium levels of HBeAg from infected cells ($n = 3$ biological replicates). S:CO signal-to-cutoff ratio, Dpi day post infection. Data are means ± SD. Statistical significance in **b**, **j**, **k**, **m**, and **n** is determined by a two-sided unpaired t-test. Source data for **b**–**n** are provided as a Source Data file.

by limiting dilution plating in 96-well plates. The sequences of gRNAs used for CRISPR knockout are as follow: *Atg5*: 5′-AACTTGTTTCACGC-TATATC-3′; *NDP52*: 5′-CAATCCAATCCTTTCGACGA-3′ HepAD38 *NDP52* stable knockdown cell line, Huh7 *NDP52* stable knockdown cell line, HepG2-NTCP *Rab9* and *Galectin-8* stable knockdown cell line were generated by the delivery of cassettes expressing hairpin RNA using pLKO.1 puro vector (Addgene #8453)[55] and selection with 2 μg ml$^{-1}$ puromycin. The sequences of hairpin RNAs used for knockdown are as follow: *NDP52*: 5′-CCGG-GAGCTGCTTCAACTGAAAGAA-CTCGAG-TTCTTTCAGTTGAAGCAGCTC-TTTTTTG-3′ *Rab9*: 5′-CCGG-AGATT GTTGATGCATTCTAAC-CTCGAG-GTTAGAATGCATCAACAATCT-TTTT TTGAAT-3′ *Galectin-8*: 5′-CCGG-CGCCTGAATATTAAAGCATTT-CTCGA G-AAATGCTTTAATATTCAGGCG-TTTTTG-3′. Reconstituted NDP 52$^{HepG2-NTCPKO}$-NDP52 cells, NDP52$^{HepG2-NTCPKO}$-ΔGIR NDP52 cells, NDP52$^{HepG2-NTCPKO}$-NDP52$_{Y97A}$ cells, NDP52$^{HepG2-NTCPKO}$-NDP52$_{A119Q}$ cells, NDP52$^{HepG2-NTCPKO}$-vector cells, NDP52$^{HepG2-NTCPWT}$ overexpressing NDP52 cells (NDP52$^{HepG2-NTCPWT}$-NDP52), NDP52$^{HepG2-NTCPWT}$-vector cells, ATG5$^{HepG2-NTCPKO}$-NDP52 cells, ATG5$^{HepG2-NTCPKO}$-vector cells, Rab9$^{HepG2-NTCPKD}$-NDP52 cells, Rab9$^{HepG2-NTCPKD}$-vector cells were generated by infection of lentiviruses expressing mCherry-tagged wild-type human NDP52, NDP52 mutants (ΔGIR, NDP52$_{Y97A}$, NDP52$_{A119Q}$), or vector and selection with 2 μg ml$^{-1}$ puromycin.

The growth curve of NDP52$^{HepG2-NTCPWT}$ and NDP52$^{HepG2-NTCPKO}$ cells were obtained by seeding $2.5 \times 10^4$ cells into 12-well plates in triplicate. Cell numbers were determined everyday for a total of 5 days.

### Transfections and cell treatments

Cells were seeded in tissue culture dishes 18 h before transfection. For coimmunoprecipitation, $2 \times 10^5$ cells of Huh7 cell line were transfected with 4 μg plasmids expressing Flag-tagged L, M and S using jetPEI (Polyplus, 101000053) and harvested 48 h post-transfection. $2 \times 10^5$ HEK293T cells were co-transfected with 2 μg plasmids encoding Flag-tagged full-length NDP52 or NDP52 deletion fragments and 2 μg plasmids encoding HA-tagged M protein. For GST pulldown, 1 μg plasmids expressing GST and GSTNDP52 were cotransfected with 3 μg plasmids expressing Flag-tagged L, M, S and M mutants. For coimmunoprecipitation of NDP52, Rab9 and M, 1 μg plasmids expressing GSTNDP52 were cotransfected with 2 μg plasmids expressing Flag-tagged M and 1 μg plasmids expressing GFP-tagged Rab9. For immunofluorescence labeling, $6 \times 10^4$ Huh7 cells were transfected with 1 μg plasmids expressing GFP fusion of L, M and core proteins. For HBV replication, $2 \times 10^5$ Huh7 cells were transfected with 4 μg plasmid expressing HBV genome ayw1.3 using Lipofectamine™ 2000 (Invitrogen™, 11668019). For lentivirus production, $10^7$ HEK293T cells were seeded in 10 cm tissue culture dishes 18 h before transfection with 14 μg lentivirus vectors expressing interested proteins along with 7 μg pCMV-VSVG

and 3.5 μg pCMV-delta 8.9. Medium was changed 16 h post transfection. Supernatants were collected every 24 h for twice.

To test the effects of NDP52 on preS2 mutants, $2 \times 10^5$ NDP52$^{HepG2-NTCPWT}$ and NDP52$^{HepG2-NTCPKO}$ cells were co-transfected with 4 μg of pPreS2Δ19-26 plus 4 μg of pHBVcore(-) or pPreS2WT plus 4 μg of pHBVcore(-) using Lipofectamine™ 2000. Four days later, cells were collected and DNA was extracted. Viral replication was analyzed by qPCR.

Seven days following the removal of doxycycline, 20 mM NH$_4$Cl, 50 μM bafilomycin A1 (Wako, 023-11641), 10 μM ectoposide (MedChemExpress, HY-13629), 200 nM rapamycin and EBSS (for Earl's Balanced Salts Solution) (Sigma Aldrich, E2888) were used to treat HepAD38 cells for 2 h. Cells were harvested for further analysis.

### Virus production and infection

For HBV production, HepAD38 cells were cultured in Dulbecco's Modified Eagle Medium (DMEM) supplemented with MEM non-essential amino acids (NEAA) (Gibco, 11140−035), 10% FBS, 100 units ml$^{-1}$ penicillin and 100 μg ml$^{-1}$ streptomycin, 5 μg ml$^{-1}$ insulin, 25 μg ml$^{-1}$ hydrocortisone (Cayman, 18226-1), 2% dimethyl sulfoxide (DMSO) (Invitrogen, D2650). HBV-containing supernatants were collected, filtered (45 μm) and centrifuged at 450 rcf for 5 min to clarify from cell debris. The supernatants were mixed with 40% polyethylene glycol (PEG) 8000 (Sigma, 89510) to a final concentration of 8% PEG, and incubated overnight at 4 °C. Viral particles were collected by centrifugation at 10,000 g for 1 h at 4 °C. The viral pellets were suspended in hepatocyte maintenance medium (PMM). Virus titers were quantified by real-time PCR.

For lentivirus production, the supernatants were filtered and clarified from cell debris by brief centrifugation. The supernatants were then mixed with 40% PEG 6000 (Sigma, 81260) to a final concentration of 8% PEG and incubated overnight at 4 °C. Viral particles were collected by centrifugation at 1500 g for 30 min. The viral pellets were suspended in complete DMEM medium.

For HBV infection experiments using the stable cell lines mentioned above, selection antibiotics were withdrawn from the culture media for at least 48 h before viral infection. Cells were seeded in Opti-MEM overnight. Cells were then grown in complete DMEM for 24 h. After treatment with 2% DMSO in complete DMEM for 24 h, cells were inoculated with HBV at 100 genome equivalents (GEq)/cell in PMM containing 4% PEG 8000 at 37 °C for 16 h. Cells were washed three times with PBS and maintained in PMM.

For lentivirus infection, cells were transduced with viral particles in the presence of 8 μg ml$^{-1}$ polybrene for 12 h. Cells were washed three times with PBS and selected with appropriate antibiotics 24 h post-infection.

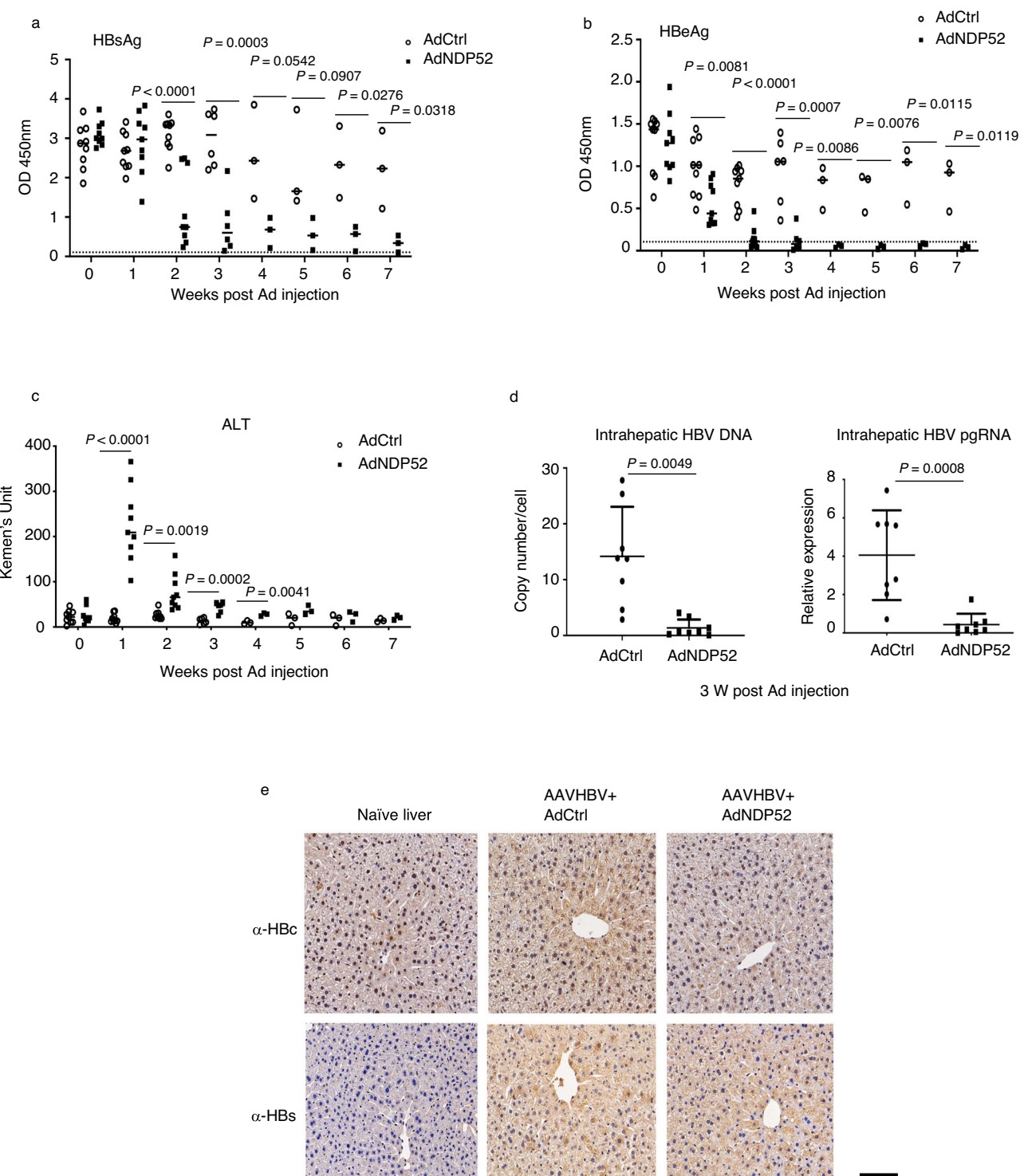

**Fig. 6 | NDP52 suppresses HBV in vivo. a–c** Mice were transduced with AAVHBV to establish chronic hepatitis B. Four weeks later (week 0), mice were injected either with adenovirus vector (AdCtrl) or expressing human NDP52 (AdNDP52). Serum levels of HBsAg (**a**), HBeAg (**b**) and alanine aminotransferase (ALT) (**c**) were measured. The dashed lines indicate the levels in naïve mice ($n = 9$ animals at weeks 0–2. $n = 6$ animals at week 3. $n = 3$ animals at weeks 4–7). **d** Quantitative PCR of HBV DNA and HBV pregenomic RNA (pgRNA) in mouse liver 3 weeks post adenovirus injection ($n = 8$ animals). Data are means ± SD. Statistical significance is determined by a two-sided unpaired t-test. **e** Immunohistochemical staining of livers with antibodies against S protein (α-HBs) or core protein (α-HBc). Source data for **a**–**d** are provided as a Source Data file.

## Southern blot analysis for viral DNA

Southern blot analysis was performed as previously described[56]. Fourteen days post HBV infection, cells were lysed with lysis buffer containing 10 mM Tris-HCl (pH 8.0), 1 mM EDTA, 1% NP-40, and 2% sucrose at 37 °C for 30 min. After removal of the nuclear pellet by centrifugation, the supernatant was incubated with 7.2% PEG 8000 containing 1.5 M NaCl at 4 °C overnight. Viral nucleocapsids were pelleted by centrifugation at 10,000 g for 30 min at 4 °C, followed by

digestion in buffer containing 0.5 mg/ml protease K, 0.5% sodium dodecyl sulfate (SDS), 150 mM NaCl, 25 mM Tris-HCl (pH 8.0), and 10 mM EDTA at 37 °C overnight. DNA released from nucleocapsid was purified by phenol-chloroform extraction, ethanol precipitation and resuspended in TE buffer (10 mM Tris-HCl, pH 8.0, 1 mM EDTA). Samples were resolved in 1% agarose gel. The gel was subjected to denaturation in 0.5 M HCl, then in a solution containing 0.5 M NaOH and 1.5 M NaCl, followed by neutralization in a buffer containing 1 M Tris-HCl (pH 7.4) and 1.5 M NaCl. DNA was then blotted onto Hybond-XL membrane (GE Healthcare) in 20x SSC (1x SSC is 0.15 M NaCl plus 0.015 M sodium citrate) buffer. HBV DNA probe labeling and hybridization were carried out by using DIG High Prime DNA Labeling and Detection Stater Kit II (Roche, 11585614910).

### Isolation and analysis of viral DNA by qPCR

HBV progeny DNA was extracted from culture supernatants using the TIANamp Virus DNA/RNA Kit (Tiangen, DP315). Intracellular HBV DNA was extracted using QIAamp DNA mini kit (Qiagen, 51306). Samples were quantified by qPCR using SYBR Green Master Mix (Yeasen, 11203ES03). The primer pairs used were: HBV rcDNA forward 5′-ATCCTGCTG CTATGCCTCATCTT −3′ and reverse 5′- ACAGTGGGG-GAAAGCCCTACGAA −3′; HBV cccDNA forward 5′-TGCACTTCGCTT-CACCT-3′ and reverse: 5′-AGGGGCATTTGGTGGTC-3′[57].

### HBV attachment and internalization assays

HBV attachment assay was performed as described previously[58,59]. NDP52$^{HepG2-NTCPWT}$ and NDP52$^{HepG2-NTCPKO}$ cells were seeded in 24-well plate and pre-treated as infection experiment. Cells were exposed to HBV particles in PMM with 4% PEG 8000 at 4 °C for 2 h. Cells were washed with PBS and collected for DNA extraction. DNA was extracted with QIAamp DNA mini kit (Qiagen, 51306) and quantified by qPCR.

HBV internalization assay was performed according to the previous description with modifications[60]. HepG2-NTCP cells were seeded in 24-well plate and pre-treated as infection experiment. Cells were exposed to HBV particles in PMM with 4% PEG 8000 at 4 °C for 2 h. Then, cells were washed with PBS and cultured at 37 °C for 16 h to allow viral internalization into cells. Cells were then trypsinized to digest the cell surface HBV and extensively washed with PBS. DNA was extracted with QIAamp DNA mini kit (Qiagen, 51306), and quantified by qPCR.

### Analysis of gene expression using qRT-PCT

Total RNA was isolated from cells or liver tissues with TRIzol reagent (Invitrogen, 15596026) and reverse transcribed to cDNA with Hifair III 1st Strand cDNA Synthesis SuperMix for qPCR (gDNA digester plus) (Yeasen, 11141ES10). The primers used were: HBV pregenomic RNA HBV2268F: 5′-GAGTGTGGATTCGCACTCC-3′, HBV2372R: 5′-GAGGC-GAGGGAGTTCTTCT-3′;[57].

### Protein analysis

For Western blot analysis, cells were lysed in NP-40 lysis buffer (50 mM Tris-HCl, pH 8.0, 150 mM NaCl, 1 mM EDTA, 1% NP-40) supplemented with protease (Roche, 04693132001) and phosphatase inhibitors (Roche, 4906845001). After boiled, protein samples were subjected to SDS-polyacrylamide gel electrophoresis and immunoblotting. The blots were blocked with 5% dry milk in Tris-buffered saline containing 0.1% Tween 20 before being probed with following primary antibody: GST (Sigma Aldrich, G7781), Flag (Sigma, F7425), HA (Sigma Aldrich, H6908), GAPDH (Abmart, M20006L), GFP (Sigma Aldrich, G1546), NDP52 (Abcam, ab68588), mcherry (ABclonal, AE002), HBV preS2 (Abcam, ab8635), HBV HBs (Thermo Fisher, PA1-73083), ATG5 (Cell Signaling Technology, 12994S), Rab9A (Cell Signaling Technology, 5118), Lamp2 (Santa Cruz, sc-18822), Galectin 8 (Santa Cruz, sc-377133). Primary antibodies were diluted in Primary Antibody Dilution Buffer (Beyotime, P0023A). Corresponding secondary antibodies (Beyotime, A0208, A0216 or Proteintech, SA00001-13) were used after washing

the membrane with TBST. Immunoreactive bands were visualized by an enhanced chemiluminescence system (Beyotime, P0018FM). The levels of protein expression were quantified with ImageJ software.

Lysosomes were isolated using Lysosome Extraction Kit (BestBio, BB-3603). Cells were collected by centrifugation at 4 °C for 5 min. Cells were washed with cold PBS, incubated on ice with Buffer A for 10 min and disrupted with Dounce homogenizer. The solution was centrifuged at $1000 \times g$ at 4 °C for 5 min. The supernatant was collected and centrifuged at $3000 \times g$ at 4 °C for 10 min. The supernatant was then centrifuged at $5000 \times g$ at 4 °C for 10 min. The supernatant was centrifuged at $25,000 \times g$ at 4 °C for 20 min. The pellet was collected and incubated on ice with Buffer B. The solution was centrifuged at $25,000 \times g$ at 4 °C for 20 min. The pellet was resolved in Buffer C and used for WB.

For co-immunoprecipitation, cells were plated in 6-well plate or 6-cm dishes and transfected with the indicated plasmids. 48 h after transfection, cells were lysed in NP-40 lysis buffer. The cell lysate was centrifuged for 10 min at 4 °C. Targeted proteins were either precipitated by antibodies or GST beads (GE, 17-0756-01), Flag beads (Millipore, A2220) or GFP beads (AlpaLife, KTSM1301) at 4 °C for 8 h. After washes in NP-40 lysis buffer, the precipitates were subjected to WB analysis.

For immunofluorescence, cells seeded on cover slips were fixed with 4% formaldehyde for 30 min at RT. For immunofluorescence of endoplasmic reticulum (ER), cells seeded on cover slips were fixed with 3% formaldehyde-0.1% glutaraldehyde for 30 min at RT. After washed with PBS, cells were permeabilized and blocked with 0.1% Triton X-100, 1% BSA in PBS for 1 h at room temperature. Cells were incubated with 0.1% Triton X-100, 1% BSA, 5% goat serum in PBS supplemented with primary antibody (1:250) at 4 °C overnight. Cells were then washed with PBS 3 times, and incubated with Alexa 488-, 594- or 647-conjugated secondary antibodies (Thermo Fisher Scientific). Nuclei were stained with 6-diamidino-2-phenylindole. Cells were visualized under 100× oil objective on Olympus confocal microscope. Colocalization Pearson's values were quantified with ImageJ. Primary antibodies are: HBs (Abcam, ab32914), calcoco2 (Santa Cruz, sc-376540), NDP52 (Abcam, ab68588), LC3A/B (Cell Signaling Technology, 12741S), HBV preS2 (Abcam, ab8635), Rab9A (Cell Signaling Technology, 5118), Lamp2 (Santa Cruz, sc-18822). The reagents used for colocalization of HBs and NDP52 in ER, Golgi or multivesicular body (MVB) are: HBsAg antibody (HB3) conjugated with Alexa Fluor™ 594 (Novus Biologicals NB500-474AF594), ER Staining Kit - Green Fluorescence – Cytopainter (Abcam, ab139481), TGN46 antibody (Abcam, ab50595), CD63 antibody (E-12) (Santa Cruz, sc-365604).

For immunohistochemistry, PFA-fixed paraffin-embedded tissue sections were dewaxed in xylene and unmasked in a citric acid solution. After blocking with normal horse serum, sections were incubated with primary antibodies against core protein (Dako, B0586) and HBs (Abcam, ab32914). Endogenous peroxidase activity was blocked by incubating the sections with 3% hydrogen peroxide. The sections were then incubated with secondary antibody (Servicebio, GB23303). The peroxidase reaction was developed. Nuclei were counterstained with hematoxylin. H&E staining was performed on deparaffinized sections with eosin and Mayer's hemalum.

**Ethical approval.** All animal experiments were performed in accordance with institutional guidelines and approved by the Institutional Animal Care and Use Committee of the Institut Pasteur of Shanghai, Chinese Academy of Sciences (Review Board Number: A2020035). All the mice were kept in a specific pathogen-free facility under biosafety levels following institutional guidelines.

**Animal experiments.** AAVHBV vector has been described previously[37]. Virus stocks were produced and titrated by the Centre de Production de Vecteurs (UMR 1089) in Nantes, France. Ad plasmid pAdHu5 deleted for E1 and E3 genes was used to generate recombinant Ad vector expressing

human NDP52. The plasmid was linearized by digestion with *PacI* and transfected into HEK293 cells to produce recombinant adenoviruses. The recombinant adenoviruses were then amplified in HEK293 cells and purified by cesium chloride gradient centrifugation.

To establish chronic HBV infection, male *C57BL/6* mice at 6-week-old were injected with $5 \times 10^{10}$ vg of AAV-HBV in 200 μl PBS via tail vein. Four weeks later, mice were selected to obtain groups with similar HBsAg and HBeAg levels. These mice and age- and sex-matched naive C57BL/6 mice were injected with either $5 \times 10^{10}$ Ad-NDP52 or Ad-empty viral particles. Animals were bled weekly. Serum HBsAg and HBeAg levels were quantified using Shanghai Kehua, ELISA HBV Test Kit. ALT in mouse serum was determined by Alanine Aminotransferase Assay Kit (Nanjing Jiancheng Bioengineering Institute, C009-2-1).

Splenocytes and lymphocytes from liver draining lymph nodes were isolated 2 and 3 weeks post recombinant adenovirus injection. Activated CD4$^+$ T cells were stained using following antibody: CD4 monoclonal antibody (GK1.5), APC-Cyanine7 (ThermoFisher, A15384), CD8α monoclonal antibody (5H10) PerCP/Cy5.5 (Invitrogen, MCD0831), CD44 Monoclonal Antibody (IM7) FITC (Invitrogen, 11-0441-82), CD62L MEL-14 APC (eBioscience, 17-0621-82).

**Statistics and reproducibility.** Statistical analyses were performed using GraphPad Prism. Two-sided unpaired *t* test was used. Data are presented as the mean and standard deviation (SD) of the mean. Co-localization was analyzed using Pearson's correlation coefficient. *P* values < 0.05 were considered significant.

All the experiments were repeated at least two times.

### Reporting summary
Further information on research design is available in the Nature Portfolio Reporting Summary linked to this article.

## Data availability
All data that support the findings of the study are available in the manuscript and the supplementary information files. Source data are provided with this paper.

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

## Acknowledgements

The authors thank Christine Neuveut and Qiang Deng for critical comments on the manuscript, Marie-Annick Buendia, Pascal Pineau, Agnès Marchio, Thomas Wollert, Jamila Faivre, Philippe Roingeard, Camille Sureau, Hugues De Rocquigny, Fabrizio Mammano for discussions and input, Pauline Verlhac, Chaolun Liu, Yimin Tong for technical help. This work was supported by Institut Pasteur grants PTR20-16 (Y.W.) and ACIP N°318 (Y.W.), National Natural Science Foundation of China N°81741070 (Y.W.), Science and Technology Commission of Shanghai Municipality N°18ZR1444000 (Y.W.).

## Author contributions

S.C. and Y.W. conceived and designed the experiments. S.C., T.X., J.Zhao., X.R., T.W., M.K., X.G., L.H., J.G., A.D., F.L. PO.V. performed the experiments. JM.C., J.Zhong., L.P., F.T., PO.V., D.Z., Y.J., M.F. contributed reagents, materials and analysis tools for experiments. S.C., T.X., M.F. and Y.W. analyzed the data. S.C. and Y.W. wrote the paper.

## Competing interests

The authors declare no competing interests.
