## [Peer Review File · Nature Communications]

NDP52 mediates an antiviral response to hepatitis B virus infection through Rab9-dependent lysosomal degradation pathwayREVIEWER COMMENTS

Reviewer #1 (Remarks to the Author):

In this paper, the author analyzed the regulatory mechanism of hepatitis B virus and obtained the following findings. (1) NDP52 shows the antiviral response to hepatitis B virus infection, (2) NDP52 binds to the HBV envelope protein and leads to its degradation, and (3) the degradation process is an Atg5-independent, Rab9-dependent alternative autophagy.

These results are unique and interesting. However, given the biology of HBV in the host, where few steps expose the envelope to the cytoplasm, it is critical to define when, where, and to what extent NDP52 binds to the HBV envelope protein in the cell to demonstrate the correctness of the conclusions in this paper. At this time, no experiments have been conducted to clarify this point, and there are doubts about the credibility of the conclusions. There are also issues regarding the quality of the data and interpretation of the results with respect to several data sets. Therefore, the manuscript is not suitable for Nat. Commn. at least in this current form.

1, In HBV infection, it is known that the Envelope Protein is quickly removed. In the proliferation process, the Envelope and Core Protein are produced separately, and the Envelope is found inside the endoplasmic reticulum, Golgi apparatus, secretory vesicles, and MVBs. In addition, fusion of the Envelope and Core Protein takes place in the MVB. In this HBV lifecycle, where, how, and to what extent NDP52, which localizes to the cytoplasm, recognizes Envelope is of critical importance, and it is essential to identify the intracellular location of NDP52 interaction with Envelope Protein.

2, In addition to NDP52, other adaptor proteins involved in autophagy are known, such as NBR1 and Tax1BP1, and it should be clarified whether NDP52 alone or other adaptor molecules have similar functions.

3, In Western Blot in Fig. 3, They observe binding of NDP52 and HBV. For further proof, there is a need for additional analysis of the colocalization of NDP52 Δ GIR Mutant and PreS2 in immunostaining. Similarly, there is a need for additional analysis of the colocalization of NDP52 with M Δ 26 and Δ 19.

4, Fig5F shows association between NDP52 and Rab9, but not enough, and IP of NDP52 and Rab9 would be recommended.

5, Experiments are needed to analyze the effects of HBV infection using NDP52-deficient mice.

Reviewer #2 (Remarks to the Author):

In their manuscript entitled " NDP52 mediates an antiviral response to hepatitis B virus infection through Rab9-dependent lysosomal degradation pathway "Shuzui Chu and colleagues describe NDP52 as a novel binding partner of LHBs/MHBs. Based on a more detailed analysis they identify aa 13-19 in the PreS2 domain as a crucial sequence of the PreS2-mediated interaction with NDP52. Moreover they observe that NDP52 exerts with respect to HBV an antiviral effect by targeting LHBs and MHBs to the lysosome. For this process the interaction with Rab9 is essential. Although this is a very interesting story there are still some open points which should be addressed.

Specific points:

Fig. 1a) Due to the posttranslational translocation of the PreS1/PreS2 domain in case of LHBs it is assumed that the N-glycosylation site at Asn4 in the PreS2 domain is not used. The schematic representation should be modified.

Fig. 1 The Co-IP experiments should be corroborated by the respective experiments using HBV expressing or even better infected cells.

To clarify the relevance of Rab9 for the interaction purified PreS2 domain and purified NDP52 should be used to study a potential direct interaction.

In light of the differences between LHBs and MHBs with respect of the membrane topology it is astonishing that there are no significant differences with respect to the interaction with NDP52.

The authors should comment on this. Moreover they should include a graphical abstract. This should include the potential mode of action. If I understand correctly the interaction with NDP52 should occur on the cytoplasmic face of the ER---how is this possible in case of MHBs. Here almost all of the PreS2 domain faces the lumen of the ER?

Fig. 2: To study the potential relevance of NDP52 for the viral entry process, the authors should include classic attachment and entry assays.

Fig. 4. Filaments and virions share the same release pathway. What about the interaction of NDP2 with filaments and their targeting to lysosomes. This could be easily studied using a core deficient genome.

Fig.6. If the authors have data describing a direct impact of NDP52 on the presentation of HBV-specific peptides on the cell surface they should include it, if not –it would be a complete new story.

Recommendation: accept after revision

Reviewer #3 (Remarks to the Author):

In this manuscript, the authors presented data to show that NDP52 could bind to the preS2 domain of the HBV envelope proteins (HBsAg) and mediate their degradation in lysosomes in a Rab9-dependent manner. They further showed that NDP52 inhibited HBV both at the entry step and the late viral replication step, and this inhibitory effect was independent of ATG5.

The experiments described in this manuscript were mostly well conducted and the results demonstrating that NDP52 could bind to the preS2 domain of HBsAg to mediate their lysosomal degradation in a Rab9-dependent manner are convincing. The authors might have discovered a novel pathway that led to the lysosomal degradation of the HBV envelope proteins. While this finding is interesting, there are some issues that are conceptually difficult to understand. In addition, additional control experiments will be required to support the conclusion of the authors.

1. The authors demonstrated that NDP52 could bind to the preS2-containing L and M HBsAg proteins. While it is possible for the cytosolic NDP52 to bind to L HBsAg, which may display its preS2 domain in the cytosol, it is difficult to understand how NDP52 could bind to M HBsAg, which displays its preS2 domain only in the ER lumen. The authors suggested that NDP52 might bind to M HBsAg in permeated or damaged endosomes and ER-Golgi membranes. It does not appear likely that there will be a large fraction of leaky membranes in the cell that allows NDP52 to enter. An alternative explanation is that the fusion protein that the authors used for the expression of M HBsAg might have caused the alteration of its membrane topology and allowed NDP52 to bind to it. In other words, the observation that NDP52 could bind to M HBsAg might be an artifact of the expression system. The results shown in Fig. 1d might represent only the interaction between NDP52 and L HBsAg. These issues should be discussed.
2. It is difficult to understand why the silencing of NDP52, which increased the level of L HBsAg and the release of HBV particles, would increase the intracellular HBV DNA level in HepAD38 cells (extended data Fig. 2).
3. It is imperative that the authors test the effect of NDP52 on the natural HBV preS2 mutant. A negative result would support the conclusion of the authors that NDP52 suppressed HBV replication via its interaction with the preS2 domain of HBsAg.
4. In Fig. 1l, the staining pattern of core was curious, as it displayed only cytoplasmic punctate staining pattern with no nuclear staining. A better control will be the small HBsAg.

5. The Southern-blot results shown by the authors are atypical, which is worrisome. The major HBV DNA bands detected should be the relaxed circular DNA and the single-stranded DNA with a minor proportion of cccDNA. The atypical Southern-blot results raised questions regarding whether the DNA bands detected by the authors were indeed HBV DNA bands or some unknown contaminants. It will be necessary for the authors to include HepAD38 cells in Figure 2b to serve as a positive control for the characterization of the DNA bands that they identified.
6. In Fig. 4b, the authors should explain why NDP52 silencing did not appear to increase the level of HBsAg in HepAD38 cells.
7. In Fig. 4c, the electrophoretic mobility between L HBsAg and M HBsAg is very different. Thus, it is inappropriate to show one protein band and indicate that it was M/L HBsAg. Instead of using anti-preS2 antibody for the western blot, the authors should instead use the anti-HBsAg antibody for the western blot and show the entire spectrum of HBsAg protein bands. The information regarding whether NDP52 differentially affects L, M and S HBsAg levels is important. This concern is also true for Fig. 4d, 5f and 5g.
8. The authors should provide detailed information to explain how they purified lysosomes (e.g., Fig. 4d).
9. The authors should explain why only one of the three cells shown in Fig. 5a, left panel, displayed the colocalization of HBsAg, NDP52 and Rab9 whereas such colocalization was inapparent in the other two cells.
10. The authors should include error bars and statistical analysis for the histograms shown in Fig. 5f and 5g.
11. The authors showed that Rab9 silencing enhanced HBV replication (Fig. 5i-k). However, a previous study indicated that Rab9 silencing had no effect on HBV (PMID: 31118260). The authors should discuss this discrepancy of the results.
12. The Fig. 6 results left more to be desired. The authors should include the HBV preS2 mutant as a control in the study. The large increase of the ALT level at one week after AdNDP52 injection might be due to the over-expression of human NDP52 and the host immune response to this protein. A control injection of AdNDP52 in the absence of HBV should help to resolve this question. This control should also be included in extended data Fig. 6. It is also desirable to test whether the activated T cells shown in extended data Fig. 6b were specific to HBV by stimulating them with HBV peptides.
13. The method used by the authors to analyze pgRNA could not distinguish pgRNA from precore RNA (e.g., Fig. 2f).
14. The authors should provide the information on how many days post-infection that they lysed HepG2-NTCP cells for Southern-blot and DNA quantification analyses.
15. As the lysosomal degradation of HBsAg was independent of ATG5 and there was no colocalization of HBsAg, NDP52 and LC3, the pathway led to the degradation of HBsAg does not involve macroautophagy and likely resembles microautophagy. This issue should be discussed.
16. In p.3, second paragraph, the authors indicated that HBV nucleocapsids bound to ER-bound envelope proteins for secretion. This statement is debatable, as it has been suggested that the envelopment of HBV core particles is associated with multivesicular bodies.

Dear Reviewers,

We have performed new experiments to answer the questions and incorporated the results in the revised manuscript.

The new figures include

- Fig. 1c lower panel, 1d lower panel, 1e, GFP-labelled S replaces core, 1g
- Fig. 3h, 3i, 3j, 3k
- Fig. 4a, lower panel
- Extended Data Fig. 1b
- Extended Data Fig. 2h
- Extended Data Fig. 6b. Extended Data Fig. 6b in the previous version is changed to Extended Data Fig. 6c.

All the changes in the text are in red.

Following are point-by-point answers.

Reviewer #1 (Remarks to the Author):

In this paper, the author analyzed the regulatory mechanism of hepatitis B virus and obtained the following findings. (1) NDP52 shows the antiviral response to hepatitis B virus infection, (2) NDP52 binds to the HBV envelope protein and leads to its degradation, and (3) the degradation process is an Atg5-independent, Rab9-dependent alternative autophagy. These results are unique and interesting. However, given the biology of HBV in the host, where few steps expose the envelope to the cytoplasm, it is critical to define when, where, and to what extent NDP52 binds to the HBV envelope protein in the cell to demonstrate the correctness of the conclusions in this paper. At this time, no experiments have been conducted to clarify this point, and there are doubts about the credibility of the conclusions. There are also issues regarding the quality of the data and interpretation of the results with respect to several data sets. Therefore, the manuscript is not suitable for Nat. Commn. at least in this current form.

1, In HBV infection, it is known that the Envelope Protein is quickly removed. In the proliferation process, the Envelope and Core Protein are produced separately, and the Envelope is found inside the endoplasmic reticulum, Golgi apparatus, secretory vesicles, and MVBs. In addition, fusion of the Envelope and Core Protein takes place in the MVB. In this HBV life cycle, where, how, and to what extent NDP52, which localizes to the cytoplasm, recognizes Envelope is of critical importance, and it is essential to identify the intracellular location of NDP52 interaction with Envelope Protein.

We have performed immunofluorescence on HepAD38 cells replicating HBV with anti-HBs and anti-NDP52 antibodies and reagents labelling ER (ER-Tracker), Golgi (anti-TGN46 antibody) and MVB (anti-CD63 antibody) and present the results in Fig. 1g. The colocalization signals of HBs and NDP52 were detected in ER, Golgi and MVB, suggesting that the recognition of the viral envelope proteins by NDP52 takes place in various organelles. In addition, we show that NDP52 affects HBV 16 h post infection (Fig. 2f), suggesting that NDP52 may recognize the preS2 domain in the endosome. Our data also suggest both subviral particle- and virion-associated L and M can be targeted by NDP52. We conclude that NDP52 specifically recognizes HBV envelope proteins through interaction with the preS2 region in various organelles and at both early and late steps of viral replication.

The topology of HBV envelope proteins and their relationship with host factors during viral life cycle are very important questions that are not fully elaborated. In the literature, few host-pathogen interactions involve HBV envelope proteins. We are aware that we do not possess all the tools to fully address the question of the mechanisms of the interaction between NDP52 and the envelope proteins. We speculate that in the delivery process of viral envelope proteins, the preS2 domain may be exposed to NDP52-containing cytoplasmic compartments to allow the interaction with it. With the advancement of technology and deeper understanding of HBV life cycle, future studies would shed more insights into the targeting of the envelope proteins by host factors. We discuss this point in the Discussion Section.

2, In addition to NDP52, other adaptor proteins involved in autophagy are known, such as NBR1 and Tax1BP1, and it should be clarified whether NDP52 alone or other adaptor molecules have similar functions.

Whether other autophagy receptors in the p62/SQSTM1-like receptor (SLR) group is involved in HBV regulation is an interesting question. We tested the effects of NBR1 and OPTN in HBV life cycle and did not observe marked phenotypes. We found that the regulatory functions of NDP52 on HBV are independent of NBR1 and OPTN. As the relationship of SLR receptors in HBV regulation is not the scope of this study, we choose not to present these results.

3, In Western Blot in Fig. 3, They observe binding of NDP52 and HBV. For further proof, there is a need for additional analysis of the colocalization of NDP52 Δ GIR Mutant and PreS2 in immunostaining. Similarly, there is a need for additional analysis of the colocalization of NDP52 with M Δ 26 and Δ 19.

New experiments have been performed to examine the colocalization of NDP52 Δ GIR mutant with M or NDP52 with M Δ 26 and Δ 19 by immunofluorescence. The results are presented in Fig. 3h,i, j. As expected, deletion of the GIR domain significantly diminishes the colocalization of NDP52 with M. Deletion of the N-terminal 26 aa in M severely affects the colocalization with NDP52, whereas M Δ 19 mutant has decreased colocalization with NDP52, which is in agreement with the results of coimmunoprecipitation assays (Fig. 3g).

4, Fig5F shows association between NDP52 and Rab9, but not enough, and IP of NDP52 and Rab9 would be recommended.

Coimmunoprecipitation of NDP52 and Rab9 has been shown in Fig. 5c. Furthermore, we showed in Fig. 5e that Rab9 immunoprecipitated NDP52 only in the presence of M.

5, Experiments are needed to analyze the effects of HBV infection using NDP52-deficient mice.

NDP52 is truncated in mouse. Murine NDP52 contains only the N-terminal 331 aa, lacking the GIR domain. We show in this study that NDP52 uses the GIR domain to recognize HBV. Thus NDP52-deficient mice may not be suitable to analyze the effects of NDP52 on HBV infection *in vivo*.

Reviewer #2 (Remarks to the Author):

In their manuscript entitled “NDP52 mediates an antiviral response to hepatitis B virus infection

through Rab9-dependent lysosomal degradation pathway “Shuzui Chu and colleagues describe NDP52 as a novel binding partner of LHBs/MHBs. Based on a more detailed analysis they identify aa 13-19 in the PreS2 domain as a crucial sequence of the PreS2-mediated interaction with NDP52. Moreover they observe that NDP52 exerts with respect to HBV an antiviral effect by targeting LHBs and MHBs to the lysosome. For this process the interaction with Rab9 is essential. Although this is a very interesting story there are still some open points which should be addressed.

Specific points:

Fig. 1a) *Due to the posttranslational translocation of the PreS1/PreS2 domain in case of LHBs it is assumed that the N-glycosylation site at Asn4 in the PreS2 domain is not used. The schematic representation should be modified.*

We have modified the symbol of the N-glycosylation site at Asn4 in Fig. 1a and Fig. 3g.

Fig. 1 *The Co-IP experiments should be corroborated by the respective experiments using HBV expressing or even better infected cells.*

We have performed new co-IP experiments in HepAD38 cells replicating HBV (cultured in the absence of doxycycline) and present the results in Fig. 1c (lower panel). The results confirm that L and M, but not S interact with NDP52.

To clarify the relevance of Rab9 for the interaction purified PreS2 domain and purified NDP52 should be used to study a potential direct interaction.

We identified NDP52 interaction with the preS2 region in yeast two hybrid assays, which indicates that their interaction is direct. In Fig. 5e, we show that interaction of Rab9 and NDP52 requires the presence of M protein, thus is not direct.

In light of the differences between LHBs and MHBs with respect of the membrane topology it is astonishing that there are no significant differences with respect to the interaction with NDP52. The authors should comment on this. Moreover they should include a graphical abstract. This should include the potential mode of action. If I understand correctly the interaction with NDP52 should occur on the cytoplasmic face of the ER---how is this possible in case of MHBs. Here almost all of the PreS2 domain faces the lumen of the ER?

As viral envelope proteins can localize in different organelles during viral replication, we have performed immunofluorescence of HepAD38 cells replicating HBV with anti-HBs and anti-NDP52 antibodies and reagents labelling ER (ER-Tracker), Golgi (anti-TGN46 antibody) and MVB (anti-CD63 antibody) and present the results in Fig. 1g. The colocalization signals of HBs and NDP52 were detected in ER, Golgi and MVB, suggesting that the recognition of the envelop proteins by NDP52 takes place in various organelles.

It is, indeed, intriguing that similar colocalization signals were observed between L and M with NDP52 (Fig. 1e,f). We speculate that in the delivery process of viral envelop proteins, the preS2 domain in L and M may be exposed to NDP52-containing cytoplasmic compartments to allow the interaction with it. The topology of HBV envelope proteins and their relationship with host factors during viral life cycle are very important questions that are not fully elaborated. In the literature, few host-pathogen interactions involve HBV envelope proteins. We are aware that we do not possess all the tools to fully address the question of the mechanisms of the

interaction between NDP52 and the envelope proteins. With the advancement of technology and deeper understanding of HBV life cycle, future studies would shed more insights into the targeting of the envelope proteins by host factors. We discuss this point in the Discussion Section.

Fig. 2: To study the potential relevance of NDP52 for the viral entry process, the authors should include classic attachment and entry assays.

We have performed attachment assays in NDP52^{HepG2-NTCPWT} and NDP52^{HepG2-NTCPKO} cells as described previously ^{1,2} and present the results in Extended Data Fig. 2h. No significant difference was observed in NDP52 WT and KO cells. The entry assay (Fig. 2f, left panel) was performed according to Iwamoto et al ³. This reference is cited in the Methods section.

Fig. 4. Filaments and virions share the same release pathway. What about the interaction of NDP2 with filaments and their targeting to lysosomes. This could be easily studied using a core deficient genome.

We showed in Fig. 1c that envelope proteins in both filaments (upper panel) and virions (lower panel) can be immunoprecipitated by NDP52. NDP52 colocalizes with L and M in HepAD38 cells cultured in both absence and presence of doxycycline (Fig. 1d), indicating that NDP52 interacts with the envelope proteins independent of viral replication.

We have performed new experiments with HepAD38 cells cultured in the presence of doxycycline, which express all the viral subgenomic RNA, but do not express pregenomic RNA, thus no viral replication. The results of WB analysis with whole cell extract and lysosomal extract are presented the results in Fig. 4a, lower panel. These results show that viral envelope proteins can be detected in the lysosome. The targeting of viral envelope proteins to the lysosome can be both NDP52-dependent and NDP52-independent, since we showed that knockdown of NDP52 drastically reduces, but does not completely abolish envelope proteins in the lysosome (Fig. 4d).

Fig.6. If the authors have data describing a direct impact of NDP52 on the presentation of HBV-specific peptides on the cell surface they should include it, if not –it would be a complete new story.

We do not have data demonstrating a direct effect of NDP52 on the presentation of HBV-specific peptides on the cell surface.

References

1. Watashi, K. *et al.* Cyclosporin A and its analogs inhibit hepatitis B virus entry into cultured hepatocytes through targeting a membrane transporter, sodium taurocholate cotransporting polypeptide (NTCP). *Hepatology* **59**, 1726-1737 (2014).
2. Chakraborty, A. *et al.* Synchronised infection identifies early rate-limiting steps in the hepatitis B virus life cycle. *Cell Microbiol* **22**, e13250 (2020).
3. Iwamoto, M. *et al.* Epidermal growth factor receptor is a host-entry cofactor triggering hepatitis B virus internalization. *Proc Natl Acad Sci U S A* **116**, 8487-8492 (2019).

Reviewer #3 (Remarks to the Author):

In this manuscript, the authors presented data to show that NDP52 could bind to the preS2

domain of the HBV envelope proteins (HBsAg) and mediate their degradation in lysosomes in a Rab9-dependent manner. They further showed that NDP52 inhibited HBV both at the entry step and the late viral replication step, and this inhibitory effect was independent of ATG5. The experiments described in this manuscript were mostly well conducted and the results demonstrating that NDP52 could bind to the preS2 domain of HBsAg to mediate their lysosomal degradation in a Rab9-dependent manner are convincing. The authors might have discovered a novel pathway that led to the lysosomal degradation of the HBV envelope proteins. While this finding is interesting, there are some issues that are conceptually difficult to understand. In addition, additional control experiments will be required to support the conclusion of the authors.

1. The authors demonstrated that NDP52 could bind to the preS2-containing L and M HBsAg proteins. While it is possible for the cytosolic NDP52 to bind to L HBsAg, which may display its preS2 domain in the cytosol, it is difficult to understand how NDP52 could bind to M HBsAg, which displays its preS2 domain only in the ER lumen. The authors suggested that NDP52 might bind to M HBsAg in permeated or damaged endosomes and ER-Golgi membranes. It does not appear likely that there will be a large fraction of leaky membranes in the cell that allows NDP52 to enter. An alternative explanation is that the fusion protein that the authors used for the expression of M HBsAg might have caused the alteration of its membrane topology and allowed NDP52 to bind to it. In other words, the observation that NDP52 could bind to M HBsAg might be an artifact of the expression system. The results shown in Fig. 1d might represent only the interaction between NDP52 and L HBsAg. These issues should be discussed.

The reviewer may refer to Fig. 1c, instead of Fig. 1d.

We have performed new immunoprecipitation assays with HepAD38 cells replicating HBV using anti-NDP52 antibody and present the results in Fig. 1c, lower panel. L and M, but not S, were detected in immune complexes with NDP52, thus ruling out potential artifacts from the expression systems.

As viral envelope proteins can localize in different organelles during viral replication, we have performed immunofluorescence of HepAD38 cells replicating HBV with anti-HBs and anti-NDP52 antibodies and reagents labelling ER (ER-Tracker), Golgi (anti-TGN46 antibody) and MVB (anti-CD63 antibody) and present the results in Fig. 1g. The colocalization signals of HBs and NDP52 were detected in ER, Golgi and MVB, suggesting that the recognition of the envelope proteins by NDP52 takes place in various organelles.

It is intriguing that NDP52 interacts with M protein. We speculate that in the delivery process of viral envelop proteins, the preS2 domain in L and M may be exposed to NDP52-containing cytoplasmic compartments to allow the interaction with it. The topology of HBV envelope proteins and their relationship with host factors during viral life cycle are very important questions that are not fully elaborated. In the literature, few host-pathogen interactions involve HBV envelope proteins. We are aware that we do not possess all the tools to fully address the question of the mechanisms of the interaction between NDP52 and the envelope proteins. With the advancement of technology and deeper understanding of HBV life cycle, future studies would shed more insights into the targeting of the envelop proteins by host factors. We discuss this point in the Discussion Section.

2. It is difficult to understand why the silencing of NDP52, which increased the level of L HBsAg and the release of HBV particles, would increase the intracellular HBV DNA level in HepAD38 cells (extended data Fig. 2).

While silencing of NDP52 increases the level of L protein and the release of HBV particles, it also decreases the targeting of HBV virions to lysosomal degradation, which increases intracellular HBV DNA in HepAD38 cells. We discussed this point in the Discussion section.

3. It is imperative that the authors test the effect of NDP52 on the natural HBV preS2 mutant. A negative result would support the conclusion of the authors that NDP52 suppressed HBV replication via its interaction with the preS2 domain of HBsAg.

To address this question, we have newly constructed three plasmids based on pHBVawy1.3. In the literature, preS2 mutants observed in patients lack consensus sequences. As the aa19-26 region in preS2 domain can be found deleted in patients and is required for the binding with NDP52, we have deleted aa19-26 in preS2 in pHBVawy1.3. In consequence, the overlapping viral polymerase ORF is also deleted. This plasmid is named pPreS2Δ19-26pol(-). To construct the control plasmid pPreS2WTPol(-), we have introduced a stop codon in the pol ORF in pHBVawy1.3 to abrogate polymerase expression. To construct the vector expressing polymerase, we have replaced ATG of the core by a stop codon in pHBVawy1.3 and named the plasmid pHBVcore(-). The detailed description for the construction of the plasmids is presented in the Methods section.

We cotransfected pHBVcore(-) with either pPreS2WTPol(-) or pPreS2Δ19-26pol(-) into NDP52^{HepG2-NTCPWT} (Ctrl) and NDP52^{HepG2-NTCPKO} (KO) cells. Four days later, cells were collected to extract DNA. Viral replication was analyzed by qPCR. The results are presented in Fig. 3k. In control cells, viral replication is at significantly higher levels in cells transfected with pPreS2Δ19-26pol(-) than those with pPreS2WTPol(-). In NDP52 KO cells, however, no significant difference in replication was observed between pPreS2WTPol(-)- and pPreS2Δ19-26pol(-)-transfected cells. These results support the notion that NDP52 suppresses HBV replication via its interaction with the preS2 domain.

4. In Fig. 1e, the staining pattern of core was curious, as it displayed only cytoplasmic punctate staining pattern with no nuclear staining. A better control will be the small HBsAg.

We have replaced the core staining with S envelope protein staining as control in Fig. 1e.

5. The Southern-blot results shown by the authors are atypical, which is worrisome. The major HBV DNA bands detected should be the relaxed circular DNA and the single-stranded DNA with a minor proportion of cccDNA. The atypical Southern-blot results raised questions regarding whether the DNA bands detected by the authors were indeed HBV DNA bands or some unknown contaminants. It will be necessary for the authors to include HepAD38 cells in Figure 2b to serve as a positive control for the characterization of the DNA bands that they identified.

We present the Southern blot results of HepAD38 cells in Extended Data Fig. 1b. It shows that digestion of DNA by EcoRI linearizes viral relaxed circular DNA. We have noticed that the viral replication forms on Southern blot are related to the states of cells following HBV infection, thus can vary according to the experiments. We have mentioned this point in the legends for Fig. 2b.

6. In Fig. 4b, the authors should explain why NDP52 silencing did not appear to increase the level of HBsAg in HepAD38 cells.

We have observed the heterogeneity of cell staining in immunofluorescence, which may be explained by the state of individual cells. Nevertheless, we have replaced the immunofluorescence graph of shNDP52 in Fig. 4b with a more representative graph.

7. *In Fig. 4c, the electrophoretic mobility between L HBsAg and M HBsAg is very different. Thus, it is inappropriate to show one protein band and indicate that it was M/L HBsAg. Instead of using anti-preS2 antibody for the western blot, the authors should instead use the anti-HBsAg antibody for the western blot and show the entire spectrum of HBsAg protein bands. The information regarding whether NDP52 differentially affects L, M and S HBsAg levels is important. This concern is also true for Fig. 4d, 5f and 5g.*

We have performed two novel experiments for protein analyses with anti-HBs antibody and present the results in Fig. 1c (lower panel) and Fig. 4a lower panel, which show the entire spectrum of HBsAg protein bands. As shown in Fig. 4a lower panel, HBs antibody detects several bands in the lysosome with molecular weight superior to 25 kd. HBsAg (24-27 kd) may be degraded in the lysosome. The major band close to 35 kd was also detected by anti-preS2 antibody and is shown in the other figures of WB with lysosome extract.

We used preS2 antibody in the experiments shown in Fig. 4d, 5f, 5g to specifically reveal NDP52-interacting envelope proteins. We agree with the reviewer that it is not appropriate to indicate the band as M/L. We have deleted the indication in the figures and changed in the text M/L to envelope proteins.

8. *The authors should provide detailed information to explain how they purified lysosomes (e.g., Fig. 4d).*

The detailed information of lysosome preparation is provided in the Methods section.

9. *The authors should explain why only one of the three cells shown in Fig. 5a, left panel, displayed the colocalization of HBsAg, NDP52 and Rab9 whereas such colocalization was inapparent in the other two cells.*

We have observed that the expression levels of a specific protein can vary in cells. We observed that the expression of Rab9 in the other two cells is relatively weak, thus rendering the colocalization inapparent.

10. *The authors should include error bars and statistical analysis for the histograms shown in Fig. 5f and 5g.*

We have changed the way of the quantification presentation in Fig. 4c,e, Fig. 5f,g.

11. *The authors showed that Rab9 silencing enhanced HBV replication (Fig. 5i-k). However, a previous study indicated that Rab9 silencing had no effect on HBV (PMID: 31118260). The authors should discuss this discrepancy of the results.*

In this report (PMID: 31118260), Rab9 was not identified as HBV regulator in siRNA screening in HepG2.2.15 cells. HepG2.2.15 is a stable cell line containing four copies of HBV genome in which the virus replicates actively. The efficiency of Rab9 silencing by siRNA transfection in HepG2.2.15 was not known. In the current study, we knocked down Rab9 expression with shRNA in HepG2-NTCP cells and infected Rab9^{HepG2-NTCPKD} cells with HBV.

Rab9 expression was efficiently reduced in Rab9^{HepG2-NTCPKD} cells (Fig. 5g). We showed that knockdown of Rab9 increased HBV replication. The discrepancy may be attributed to the difference in the sensitivity of the methods. We discuss this point in the Discussion section.

12. *The Fig. 6 results left more to be desired. The authors should include the HBV preS2 mutant as a control in the study. The large increase of the ALT level at one week after AdNDP52 injection might be due to the over-expression of human NDP52 and the host immune response to this protein. A control injection of AdNDP52 in the absence of HBV should help to resolve this question. This control should also be included in extended data Fig. 6. It is also desirable to test whether the activated T cells shown in extended data Fig. 6b were specific to HBV by stimulating them with HBV peptides.*

For technical reasons, we did not include HBV preS2 mutant in the *in vivo* study. This mutant is not replication-competent by itself, because the overlapping polymerase ORF is deleted. In this case, providing viral polymerase *in trans* is required to support HBV preS2 mutant replication, which is more feasible in cell culture system than in the mouse liver.

The ALT data from naive mice injected with AdNDP52 are added in Extended Data Fig. 6b. These results show that overexpression of NDP52 augments ALT in both mice with HBV and naive mice. However, mice with HBV injected with AdNDP52 present significantly higher levels of ALT than those without HBV, suggesting that NDP52-mediated HBV degradation plays a predominant role in hepatocyte lysis.

It would be interesting to characterize the activated T cells in detail. We did not pursue this research direction, because T cell response to HBV infection is not the focus of the current study.

13. *The method used by the authors to analyze pgRNA could not distinguish pgRNA from precore RNA (e.g., Fig. 2f).*

We used the primers HBV2268F and HBV2372R for pregenomic RNA detection, which have been described by Sun et al in *Methods Mol Biol* 1540, 1-14, doi:10.1007/978-1-4939-6700-1_1 (2017) (this reference is provided in the manuscript). It is true that these primers can amplify precore RNA.

14. *The authors should provide the information on how many days post-infection that they lysed HepG2-NTCP cells for Southern-blot and DNA quantification analyses.*

We have provided this information for Southern blot analysis in the Methods section. Southern blot analysis was performed fourteen days post HBV infection. We have added the information for Southern blot and HBV DNA quantitative PCR analysis in related figure legends.

15. *As the lysosomal degradation of HBsAg was independent of ATG5 and there was no colocalization of HBsAg, NDP52 and LC3, the pathway led to the degradation of HBsAg does not involve macroautophagy and likely resembles microautophagy. This issue should be discussed.*

Microautophagy involves invaginations of the lysosomal membrane into the lysosomal lumen to take in substrates. Here we demonstrate that HBV envelopes are targeted by NDP52/Rab9 to the lysosome for degradation, which is not related to microautophagy. We did not know if viral envelope proteins are delivered to the lysosome via the double membrane

autophagosome, thus via macroautophagy. However, our observations that autophagy stimuli such as starvation, etoposide and rapamycin induce accumulation of M and L in the lysosome suggest that selective autophagy pathway may be involved in HBV degradation. This selective autophagy pathway is different from canonical autophagy pathway: it is dependent of Rab9, but independent of ATG5 and LC3. We discussed this point in the Discussion section.

16. In p.3, second paragraph, the authors indicated that HBV nucleocapsids bound to ER-bound envelope proteins for secretion. This statement is debatable, as it has been suggested that the envelopment of HBV core particles is associated with multivesicular bodies.

We have deleted the word ER-bound in the revised version.

REVIEWERS' COMMENTS

Reviewer #1 (Remarks to the Author):

The manuscript seems to be much better, responding to the comments. I thereby consider the paper worthy of publication.

Reviewer #2 (Remarks to the Author):

The authors performed a very careful revision. In the revised version all points raised in my previous review are adequately addressed. The authors made interesting and relevant observations which are clearly described and well discussed.

Reviewer #3 (Remarks to the Author):

The authors had included new data to try to address the comments of the previous review. As such, the manuscript had been improved. However, there remains some issues that require attention:

Major comments:

1. Although the authors conducted the co-IP experiment to demonstrate that M-HBsAg could be co-immunoprecipitated with NDP52, it remains problematic regarding how the cytosolic protein NDP52 binds to the luminal domain of the M-HBsAg protein. The co-IP was conducted after membranes were solubilized with NP-40. Thus, the binding to these two proteins to each other could have easily happened after cells were lysed. This could explain why the glycosylated form of the M-HBsAg was co-IP'ed with NDP52. The suggestion that the M-HBsAg might have different membrane topologies during viral replication is not convincing, and the suggestion that intracellular membrane vesicles were leaky to allow NDP52 to enter is farfetched. The authors should tone down on the possible interaction between NDP52 and M-HBsAg and focus on the interaction between NDP52 and L-HBsAg, which is known to partially display its preS2 domain in the cytosol.

2. There were misstatements in describing the pPreS2delta19-26pol(-) plasmid. The deletion of aa19-26 did not abolish the expression of pol. This deletion occurs in the spacer domain of the HBV polymerase and unlikely would affect the polymerase activity of HBV. Thus, it is incorrect to indicate that this plasmid is pol(-). The description about this plasmid in the manuscript and the statements about the need to conduct the transcomplementation experiment should be revised.

Minor comments:

3. In the last paragraph of p.9, the authors stated, "suggesting that NDP52-mediated HBV degradation plays a predominant role in hepatocyte lysis." The results shown by the authors were that human NDP52 induced liver injury in mice, which was exacerbated by HBV. The authors did not show that this liver injury was due to "NDP52-mediated HBV degradation". This sentence should be revised.

4. In p.10, in the second paragraph of Discussion, the authors stated, "NDP52 may recognize the preS2 region in the endosome". As mentioned above, whether NDP52 binds to the preS2 domain

in the cytosol or in the lumen of membranous compartment is unresolved. This overstatement should be revised.

5. In Figure 4 legend, "whole cell lysis or lysosome lysis" should be "whole cell lysates" and "lysosomal extracts".

Dear Reviewers,

We have revised our manuscript according to the comments. All the changes in the text are in red.

Following are point-by-point answers.

Reviewer #1 (Remarks to the Author):

The manuscript seems to be much better, responding to the comments. I thereby consider the paper worthy of publication.

Reviewer #2 (Remarks to the Author):

The authors performed a very careful revision. In the revised version all points raised in my previous review are adequately addressed. The authors made interesting and relevant observations which are clearly described and well discussed.

Reviewer #3 (Remarks to the Author):

The authors had included new data to try to address the comments of the previous review. As such, the manuscript had been improved. However, there remains some issues that require attention:

Major comments:

1. Although the authors conducted the co-IP experiment to demonstrate that M-HBsAg could be co-immunoprecipitated with NDP52, it remains problematic regarding how the cytosolic protein NDP52 binds to the luminal domain of the M-HBsAg protein. The co-IP was conducted after membranes were solubilized with NP-40. Thus, the binding to these two proteins to each other could have easily happened after cells were lysed. This could explain why the glycosylated form of the M-HBsAg was co-IP'ed with NDP52. The suggestion that the M-HBsAg might have different membrane topologies during viral replication is not convincing, and the suggestion that intracellular membrane vesicles were leaky to allow NDP52 to enter is farfetched. The authors should tone down on the possible interaction between NDP52 and M-HBsAg and focus on the interaction between NDP52 and L-HBsAg, which is known to partially display its preS2 domain in the cytosol.

We agree that the interaction of M with NDP52 detected by co-IP experiments can take place following cell lysis. However, we show by immunofluorescence that M can colocalize with NDP52 and deletion of interaction domains either in NDP52 (Δ GIR) or in M (Δ 26) significantly diminishes colocalization signals (Fig. 1e, 3h). Even though we do not know how the preS2 region of M is recognized by NDP52, we cannot deny these observations.

The luminal localization of the preS2 region in M protein is the currently accepted standard model of M topology, which was established in the nineties of the last century. Given the highly dynamic nature of the Dane particle and subviral particles, it is safe to assume that there are still unknown aspects in the topology of HBV envelop proteins in the course of viral life cycle, which will be progressively unveiled with the development of new technology and research tools. We discuss this point in the Discussion section.

We have deleted the membrane vesicle leaky sentence.

2. *There were misstatements in describing the pPreS2delta19-26pol(-) plasmid. The deletion of aa19-26 did not abolish the expression of pol. This deletion occurs in the spacer domain of the HBV polymerase and unlikely would affect the polymerase activity of HBV. Thus, it is incorrect to indicate that this plasmid is pol(-). The description about this plasmid in the manuscript and the statements about the need to conduct the transcomplementation experiment should be revised.*

We agree. We have changed the names of the plasmid pPreS2Δ19-26pol(-) to pPreS2Δ19-26 and accordingly pPreS2WTpol(-) to pPreS2WT. We have revised the description in the Methods section as following: The overlapping viral polymerase ORF at the amino acid positions 308-315 is also deleted in pPreS2Δ19-26 and the arginine at the position 307 is changed to isoleucine. Although the 308-315 deletion occurs in the spacer domain of the polymerase, full length polymerase was provided *in trans* to ensure efficient viral replication.

Minor comments:

3. *In the last paragraph of p.9, the authors stated, “suggesting that NDP52-mediated HBV degradation plays a predominant role in hepatocyte lysis.” The results shown by the authors were that human NDP52 induced liver injury in mice, which was exacerbated by HBV. The authors did not show that this liver injury was due to “NDP52-mediated HBV degradation”. This sentence should be revised.*

We have deleted the sentence “suggesting that NDP52-mediated HBV degradation plays a predominant role in hepatocyte lysis.” We have revised the conclusive sentence to “NDP52 may be important for HBV clearance *in vivo*.”

4. *In p.10, in the second paragraph of Discussion, the authors stated, “NDP52 may recognize the preS2 region in the endosome”. As mentioned above, whether NDP52 binds to the preS2 domain in the cytosol or in the lumen of membranous compartment is unresolved. This overstatement should be revised.*

We have revise to “NDP52 may recognize the preS2 region at the early step of viral life cycle.”

5. *In Figure 4 legend, “whole cell lysis or lysosome lysis” should be “whole cell lysates” and “lysosomal extracts”.*

We have changed to “whole cell lysate (WCL) and lysosomal lysate (LL)”.